# Disuse-associated loss of the protease LONP1 in muscle impairs mitochondrial function and causes reduced skeletal muscle mass and strength

Zhisheng Xu[1,10], Tingting Fu[1,10], Qiqi Guo[1,10], Danxia Zhou[1], Wanping Sun[1], Zheng Zhou[1], Xinyi Chen[1], Jingzi Zhang [2], Lin Liu[1], Liwei Xiao[1], Yujing Yin[1], Yuhuan Jia[1], Erkai Pang[3], Yuncong Chen [4], Xin Pan [5], Lei Fang [2], Min-sheng Zhu[6], Wenyong Fei[3], Bin Lu[7] & Zhenji Gan [1,8,9 ✉]

Mitochondrial proteolysis is an evolutionarily conserved quality-control mechanism to maintain proper mitochondrial integrity and function. However, the physiological relevance of stress-induced impaired mitochondrial protein quality remains unclear. Here, we demonstrate that LONP1, a major mitochondrial protease resides in the matrix, plays a role in controlling mitochondrial function as well as skeletal muscle mass and strength in response to muscle disuse. In humans and mice, disuse-related muscle loss is associated with decreased mitochondrial LONP1 protein. Skeletal muscle-specific ablation of LONP1 in mice resulted in impaired mitochondrial protein turnover, leading to mitochondrial dysfunction. This caused reduced muscle fiber size and strength. Mechanistically, aberrant accumulation of mitochondrial-retained protein in muscle upon loss of LONP1 induces the activation of autophagy-lysosome degradation program of muscle loss. Overexpressing a mitochondrial-retained mutant ornithine transcarbamylase (ΔOTC), a known protein degraded by LONP1, in skeletal muscle induces mitochondrial dysfunction, autophagy activation, and cause muscle loss and weakness. Thus, these findings reveal a role of LONP1-dependent mitochondrial protein quality-control in safeguarding mitochondrial function and preserving skeletal muscle mass and strength, and unravel a link between mitochondrial protein quality and muscle mass maintenance during muscle disuse.

[1] State Key Laboratory of Pharmaceutical Biotechnology and MOE Key Laboratory of Model Animal for Disease Study, Model Animal Research Center, Department of Spine Surgery, Nanjing Drum Tower Hospital, The Affiliated Hospital of Nanjing University Medical School, Nanjing University Medical School, Nanjing University, Nanjing, China. [2] Jiangsu Key Laboratory of Molecular Medicine & Chemistry and Biomedicine Innovation Center, Medical School of Nanjing University, Nanjing, China. [3] Sports Medicine Department, Northern Jiangsu People's Hospital, Clinical Medical College, Yangzhou University, Yangzhou, China. [4] State Key Laboratory of Coordination Chemistry, School of Chemistry and Chemical Engineering, Chemistry and Biomedicine Innovation Center (ChemBIC), Nanjing University, Nanjing, China. [5] State Key Laboratory of Proteomics, Institute of Basic Medical Sciences, National Center of Biomedical Analysis, Beijing, China. [6] The State Key Laboratory of Pharmaceutical Biotechnology and MOE Key Laboratory of Model Animal for Disease Study, Model Animal Research Center, Nanjing University Medical School, Nanjing University, Nanjing, China. [7] Department of Biochemistry and Molecular Biology, School of Basic Medical Sciences, Hengyang Medical School, University of South China, Hengyang, China. [8] Jiangsu Key Laboratory of Molecular Medicine, Nanjing University Medical School, Nanjing University, Nanjing, China. [9] Chemistry and Biomedicine Innovation Center (ChemBIC), Nanjing University, Nanjing, China. [10]These authors contributed equally: Zhisheng Xu, Tingting Fu, Qiqi Guo. ✉email: ganzj@nju.edu.cn

Mitochondria have diverse functions and are essential organelles that require continuous surveillance to maintain their function. Mitochondrial proteases serve as the first-line quality control by selective targeting and removal of damaged or dysfunctional mitochondrial proteins to ensure the proper functional integrity of mitochondria[1–4]. Disturbed mitochondrial proteostasis occurs under cellular stress conditions and is associated with aging and with a wide variety of human illnesses, including metabolic disorders, cancer, and neurodegenerative diseases[1,2,4,5]. However, the physiological relevance of stress-induced impaired or dysregulated mitochondrial protein quality control remains unclear.

The safeguard of a functional mitochondrial system is particularly important for skeletal muscle, the largest metabolically active and highly structured tissue that is often affected in diseases of mitochondrial dysfunction. Significant evidence suggests that mitochondrial quality-control mechanisms are essential for maintaining skeletal muscle function in response to a myriad of physiologic or pathophysiological stresses[4,6–8]. Skeletal muscle possesses a remarkable capacity to adapt its mass to changes in physical activity. Muscle disuse as in decreased activity or aging causes muscle loss and weakness. The loss of muscle mass and strength exacerbates immobility, and this can lead to a vicious cycle of further muscle loss and weakness that lowers the quality of life and increases mortality. Indeed, physical inactivity-associated skeletal muscle loss and weakness is a major health problem and is a common consequence of a variety of disease conditions, including bed rest, chronic illness, and neuromuscular disorders[4,7,9–11], whereas exercise is effective in counteracting the effects of many chronic diseases on muscle mass and function[4,12,13]. Skeletal muscle mass is controlled by a fine balance between protein synthesis and protein degradation processes triggered by changes in physical activity[11,14,15], yet the intrinsic muscular signaling networks driving muscle mass loss particularly during the early phase of muscle disuse remain unclear.

Both ATP-dependent and ATP-independent mitochondrial proteases have been identified in mammals for selective degradation of the excess non-assembled, misfolded, or damaged mitochondrial proteins[1–4]. Approximately two thirds of the 1200 mitochondrial proteins reside in the matrix and inner membrane, regulation of the enormously protein-dense matrix environment within the organelle is therefore essential for maintaining the proper function of mitochondria. ATP-dependent mitochondrial proteases Lon protease homolog (LONP1) and Clp protease proteolytic subunit (CLPP), stationed in the mitochondrial matrix, are the center of mitochondrial protein quality control[1–4]. LONP1 is an evolutionarily conserved serine peptidase that controls mitochondrial protein quality from yeast to human[1–4]. In mammals, several mitochondrial regulatory proteins such as aconitase (ACO), cytochrome c oxidase isoform (Cox4i1), steroidogenic acute regulatory protein (STAR), and transcription factor A (TFAM) have been identified as LONP1 substrates under basal or stress conditions[2,4,16,17]. The embryonic lethality of mice with deletion of the *Lonp1* gene further highlighted the essential role of LONP1 for life[18]. Mutations in LONP1 have been implicated in CODAS syndrome, a human genetic disease that causes multisystem malfunctions[19], whereas changes in the expression levels of LONP1 under cellular stress conditions have also been reported[1,2], Recently, the quality of mitochondria has emerged as important control process that regulates skeletal muscle function[20–23]. However, it has yet to be explored whether LONP1-meidated mitochondrial protein quality control plays a role in sensing skeletal muscle stress and regulating its function.

In this work, we find that disuse-related muscle loss in mice and humans is associated with decreased mitochondrial LONP1 protein. Using loss-of-function strategies in mice and primary muscle cells, we demonstrate the deleterious effects of disuse-induced loss of LONP1 on mitochondrial function as well as skeletal muscle mass and strength. Ablation of LONP1 in skeletal muscle results in impaired mitochondrial protein turnover, which alters mitochondrial function and triggers the activation of autophagy-lysosome degradation. This causes reduced muscle mass and strength. Importantly, many of these features are recapitulated in mice overexpressing a mitochondrial-retained ΔOTC, a known protein degraded by LONP1, in skeletal muscle. Our work highlights the importance of LONP1-dependent mitochondrial mechanism for skeletal muscle function, and mitochondrial protein quality and muscle mass maintenance are intimately linked.

## Results

**Disuse-related muscle loss in mice and humans is associated with decreased mitochondrial LONP1 protein.** To investigate whether mitochondrial protein quality-control mechanisms are implicated in regulating skeletal muscle remodeling upon physical inactivity, we first examined the major mitochondrial proteases in the matrix during denervation-induced muscle disuse in mice. Starting at 5 days following denervation, we observed gradual and significant decreases in the protein expression of mitochondrial protease LONP1 along with apparent reductions in gastrocnemius (GC) and tibialis anterior (TA) muscle weight (Fig. 1a, b and Supplementary Fig. 1a). Notably, we found decreased levels of mitochondrial protease CLPP and dynamic protein OPA1 only at later stages of denervation-induced muscle loss (Fig. 1a, b), whereas the levels of other mitochondrial dynamic proteins MFN1, MFN2, and DRP1 remained unchanged during the denervation process (Fig. 1a, b). These results indicate that decreased mitochondrial LONP1 protein accompanies the rapid muscle mass loss during the early phase of muscle disuse. Next, we tested if mitochondrial LONP1 is also regulated by a hindlimb immobilization, another model of muscle disuse in mice[24]. We confirmed that LONP1 protein expression decreased in skeletal muscle after immobilization-induced muscle disuse (Fig. 1c, d), correlating with significant muscle loss (Supplementary Fig. 1b), whereas no decreases in the protein levels of CLPP, OPA1, MFN1, MFN2, and DRP1 were observed during the immobilization process (Fig. 1c, d and Supplementary Fig. 1c, d).

Using a reporter mouse containing the fluorescent reporter MitoTimer, we assessed the overall protein turnover status of the mitochondrial reticulum in skeletal muscle. Consistent with reduced mitochondrial protease LONP1 expression, we found that the red:green ratio of the MitoTimer was significantly shifted towards red fluorescence, an indicator of slow mitochondrial protein turnover, in both TA and extensor digital longus (EDL) muscles at 5 days following denervation (Fig. 1e and Supplementary Fig. 1e). Similarly, significantly increased MitoTimer red:green ratio was also detected in muscles following immobilization (Fig. 1f). Thus, these data demonstrate that downregulation of the mitochondrial LONP1, along with impaired mitochondrial protein turnover, accompanies the process of disuse-induced skeletal muscle loss.

We next explored whether the downregulated mitochondrial LONP1 protein also occurs in disused muscle in humans. Supraspinatus muscle, one of the most important muscles of rotator cuff, is often injured during rotator cuff tears, causing a common and challenging orthopedic problem[25,26]. Disused supraspinatus muscles due to tendon detachment can result in progressive muscle loss, and the severity of supraspinatus muscle loss is linked to the time elapsed between the onset of rotator cuff symptoms and the delay of diagnosis[25,26]. By calculating the occupation ratio which is the ratio between the surface of the

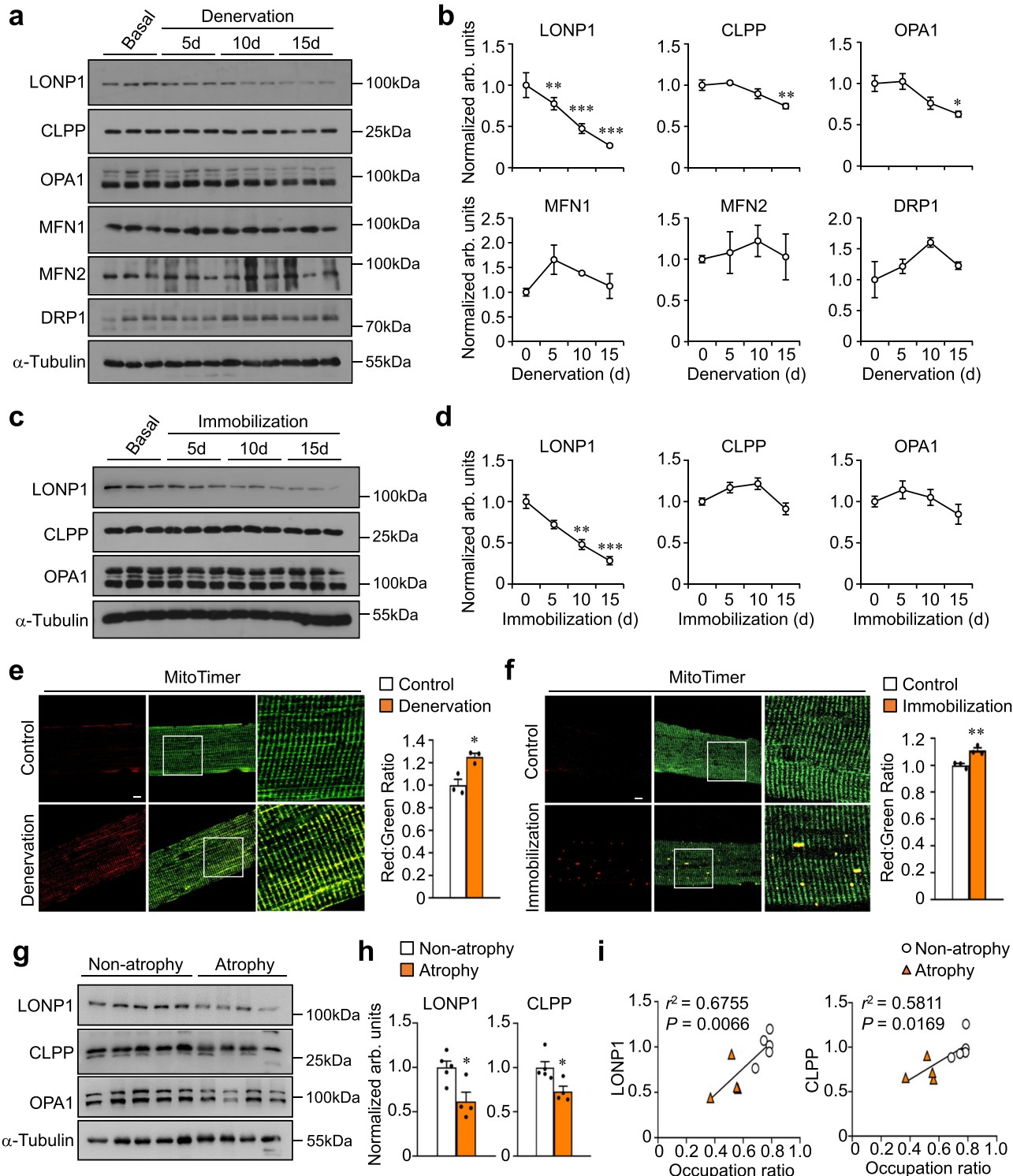

cross-section of the muscle belly and that of the supraspinatus fossa, a previous study have reported a reliable measurement of supraspinatus muscle atrophy using an MRI-based method[27]. We thus examined the regulation of mitochondrial LONP1 protein in supraspinatus muscle samples from patients with and without muscle atrophy. Based on the occupation ratio of the supraspinatus fossa by the muscle belly and previous report[27], patients were divided into two groups (non-atrophy group, mean ratio 0.76; atrophy group, mean ratio 0.50) (Supplementary Table 1). We found that the protein levels of LONP1 and CLPP were significantly reduced in human muscles from the atrophy group

compared to the non-atrophy group (Fig. 1g, h), In addition, both LONP1 and CLPP proteins exhibited a significant positive correlation with occupation ratios (Fig. 1i). In contrast, the levels of OPA1 protein were not different between atrophy and non-atrophy groups, and neither exhibited a significant correlation with occupation ratios (Fig. 1g and Supplementary Fig. 1f, g). Overall, these findings demonstrate that disuse-related muscle loss in mice and humans is associated with decreased mitochondrial proteases, and suggest that LONP1 may act as an early sensor in the control circuitry during disuse-induced muscle loss.

**Fig. 1 Disuse-related muscle loss in mice and humans is associated with decreased mitochondrial LONP1 protein. a** Representative Western blot analysis performed with gastrocnemius (GC) muscle total protein extracts prepared from non-denervated (basal) and denervated limbs of WT mice for the indicated number of days (d) using indicated antibodies. **b** Quantification of LONP1/Tubulin, CLPP/Tubulin, OPA1/Tubulin, MFN1/Tubulin, MFN2/Tubulin and DRP1/Tubulin signal ratios normalized (=1.0) to basal is shown. $n = 3$–6 mice per group. **c** Representative western blot analysis performed with GC muscle total protein extracts prepared from non-immobilized (basal) and immobilized limbs of WT mice for the indicated number of days (d) using indicated antibodies. **d** Quantification of LONP1/Tubulin, CLPP/Tubulin, and OPA1/Tubulin signal ratios normalized (=1.0) to basal is shown. $n = 3$–6 mice per group. (**e**, **f**) (Left) Representative confocal images of extensor digital longus (EDL) muscles in HSA-Cre/MitoTimer mice at 5 days post denervation (**e**) or 10 days post immobilization (**f**). The scale bar represents 10 μm. (Right) Quantification of MitoTimer Red:Green ratio normalized (=1.0) to control is shown. $n = 3$ mice per group. **g**, **h** Supraspinatus muscle samples from 9 rotator cuff tear patients who underwent rotator cuff repair surgery were used for this analysis. **g** Representative Western blot analysis of LONP1, CLPP, and OPA1 expression in human supraspinatus muscles. Based on the occupation ratio, patients were divided into two groups (non-atrophy group, mean ratio 0.76; atrophy group, mean ratio 0.50). **h** Quantification of LONP1/Tubulin and CLPP/Tubulin data shown in (**g**). $n = 4$–5 patients per group. (**i**) Correlation between the protein expression of LONP1 and CLPP to the occupation ratios. $n = 9$ patients. Pearson's correlation analysis was used to determine the correlation. Values represent mean ± SEM; for (**b**) and (**d**), *$P < 0.05$, **$P < 0.01$, ***$P < 0.001$ versus corresponding controls determined by one-way ANOVA coupled to a Fisher's LSD post-hoc test; for (**e**), (**f**), and (**h**), *$P < 0.05$, **$P < 0.01$ versus corresponding controls determined by two-tailed unpaired Student's $t$ test. Source data are provided as a Source Data file.

**Skeletal muscle-specific ablation of LONP1 impairs muscle growth and causes muscle weakness and precocious aging.** To assess the physiological consequence of disuse-induced loss of LONP1 in skeletal muscle in vivo, we generated skeletal muscle-specific *Lonp1*-knockout mice (referred to as LONP1 mKO) by breeding *Lonp1*-floxed mice (*Lonp1*^f/f^) with mice expressing Cre in postnatal skeletal muscle under the control of the human skeletal actin (*HSA*) promoter. This resulted in efficient postnatal deletion of LONP1 in skeletal muscle. Consistent with previously published data[28], efficient ablation of LONP1 mediated by HSA-Cre did not occur until ~7 days after birth (Supplementary Fig. 2a). As expected, the expression of LONP1 protein levels was markedly reduced in multiple muscle types, but not in the heart, from LONP1 mKO mice compared to WT littermates (Supplementary Fig. 2b), whereas the protein levels of CLPP, OPA1, MFN1, MFN2, and DRP1 were not reduced in LONP1 mKO muscle (Supplementary Fig. 2c). Mice with muscle-specific knockout of *Lonp1* were born at normal Mendelian ratios and were indistinguishable from their WT littermates during the first 2 weeks of postnatal life. Histological analysis of muscles revealed normal muscle architecture and absence of myofiber degeneration features such as center-nucleated fibers (Supplementary Fig. 2d). We did not detect the induction of major inflammation-associated genes in LONP1 mKO muscles (Supplementary Fig. 2e). The expression of key myogenic regulator genes including *Myf5*, *Myod1*, *Myog*, *Myf6*, *Pax3*, and *Pax7* were not different between LONP1 mKO and WT control mice (Supplementary Fig. 2f). These results indicate that ablation of *Lonp1* neither affects muscle differentiation nor development. However, LONP1 mKO mice gradually displayed an impaired muscle growth phenotype, with the muscle size difference between the LONP1 mKO mice and their WT littermates becoming more pronounced with age (Fig. 2a). Notably, we found no change in the hepatic expression of growth hormone *Igf1* or *Fos* genes (Supplementary Fig. 2g). The LONP1 mKO mice could breed successfully and showed the similar survival as respective WT littermates at least up to 1.5 years of age. At 6 weeks of age but not at 2 weeks of age, LONP1 mKO mice showed a significant reduction in muscle weight and myofiber size under basal conditions (Fig. 2b–e and Supplementary Fig. 2h), they showed more pronounced myofiber atrophy and muscle loss in GC and TA muscles in response to denervation (Fig. 2b–e). The grip strength of adult LONP1 mKO mice was significantly reduced (Fig. 2f). We also measured muscle force ex vivo, tetanic forces were reduced to less than 50% in 10-week-old LONP1 mKO EDL muscles compared to WT controls (Fig. 2g). Thus, loss of LONP1 results in impaired muscle growth and muscle weakness. We also assessed muscle exercise performance in living mice. Acute

running endurance tests revealed that LONP1 mKO mice could run for significantly shorter time and distance (~75%) compared to WT littermates (Fig. 2h). To further evaluate muscle fuel utilization during exercise, real-time respiratory exchange ratio (RER) and oxygen utilization (VO₂) were measured during a high-intensity exercise protocol. Interestingly, LONP1 mKO mice had a higher RER than WT controls during exercise, and this was accompanied by reduced exercise tolerance (Fig. 2i). Because a higher RER typically reflects a substrate shift in favor of carbohydrate metabolism, these data suggest that LONP1 mKO mice consume less fat and depend on glucose metabolism more than WT controls during exercise. This is consistent with observations that LONP1 mKO mice consumed less oxygen during the exercise period (as reflected by ΔVO₂) (Fig. 2j). In contrast, LONP1 mKO mice showed significantly higher levels (~56%) of blood lactate, a marker of increased glycolysis, following exercise than in WT controls (Fig. 2k). The reduced exercise endurance in LONP1 mKO mice was not due to reduce in slow-twitch type I muscle fibers. Indeed, surprisingly, MHC1 immunofluorescence staining revealed a striking increase in the number of type I fibers in LONP1 mKO GC muscles compared to WT controls (Supplementary Fig. 2i). Moreover, expression of the gene encoding the major slow-twitch type I myosin isoform MHC1 (*Myh7* gene) and slow-twitch troponin genes (*Tnni1*, *Tnnc1*, and *Tnnt1*) was increased in LONP1 mKO GC muscles (Supplementary Fig. 2j). In contrast, expression of the fast-twitch type II myosin isoform MHC2b (*Myh4* gene) and fast-twitch troponin genes (*Tnni2*, *Tnnc2*, and *Tnnt3*) was reduced in LONP1 mKO GC muscles (Supplementary Fig. 2j). Together, these results demonstrate a change in muscle fiber size and strength in the absence of muscle LONP1 that compromises muscle performance.

We also performed transcriptomic analysis in muscle from 6-week-old WT and LONP1 mKO mice. Ablation of LONP1 in skeletal muscle resulted in the up-regulation of 982 genes and the down-regulation of 940 genes. Interestingly, GSEA analysis of transcriptomic data revealed that genes upregulated by LONP1 deficiency were significantly enriched during normal aging in WT mice (Fig. 2l). Consistently, LONP1 mKO mice started to show kyphosis, a sign of precocious aging, at 6 months of age (Fig. 2m). Together, these results establish an essential role of LONP1 for the maintenance of skeletal muscle mass and fitness in vivo.

**Loss of LONP1 leads to severe mitochondrial structural and functional abnormalities in skeletal muscle.** To define the molecular basis of the muscle phenotype in LONP1 mKO mice, we first analyzed mitochondrial function. Mitochondrial volume density, size, and ultrastructure were assessed by electron microscopy (EM) in both the intermyofibrillar (IMF) and

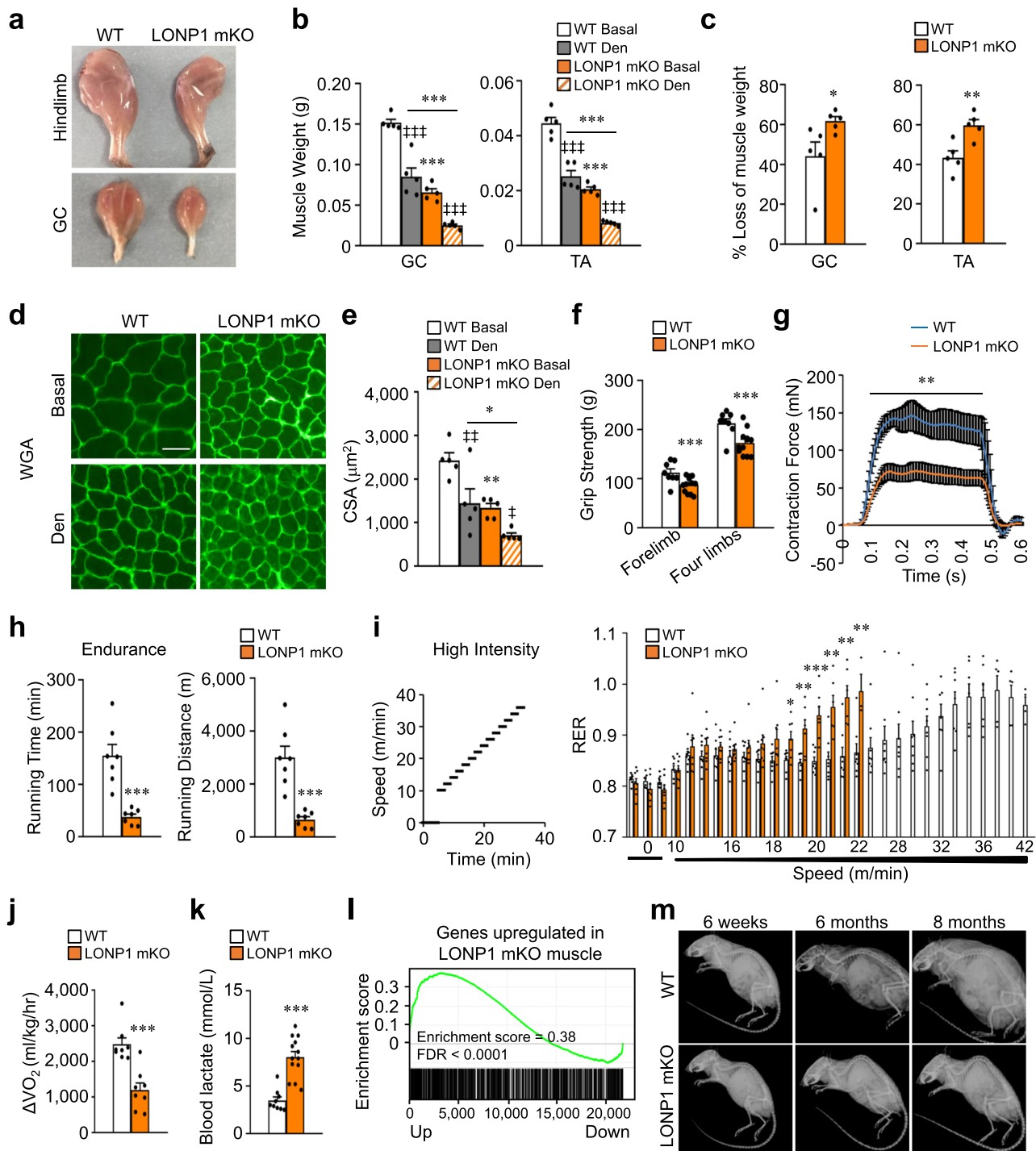

**Fig. 2 Skeletal muscle-specific ablation of LONP1 impairs muscle growth and causes muscle weakness and precocious aging. a** Representative hindlimbs and GC muscles from indicated mice at the age of 6 weeks. **b, c** GC and TA muscles weight in WT and LONP1 mKO mice in basal conditions (Basal) and after 15 days of denervation (Den). $n = 5$ mice per group. **d** WGA (green) staining of GC muscle from indicated mice. Scale bar: 50 μm. n = 5 mice per group. (**e**) Cross-sectional areas of GC myofibers. $n = 5$ mice per group. **f** LONP1 mKO mice showed decreased muscle-grip strength compared to WT controls. $n = 8$–12 mice per group. **g** Measurement of muscle tetanic contraction reveals a reduced contraction force in LONP1 mKO EDL muscle. $n = 3$–4 mice per group. **h** Bars represent mean running time and distance for 10-week-old male LONP1 mKO mice and WT controls on a motorized treadmill. $n = 7$ mice per group. **i** (Left) Schematic depicts the increments of speed over time. (Right) Respiratory exchange ratio (RER) during the course of the high-intensity exercise in indicated mice. $n = 8$ mice per group. **j** Peak $\Delta VO_2$ (increase in oxygen consumption during exercise) is graphed. $n = 8$ mice per group. **k** Bars represent mean blood lactate levels for indicated mice following a 25-min run on a motorized treadmill. $n = 9$–13 mice per group. (**l**) GSEA of genes upregulated in LONP1-deficient muscle in relation to normal aging in WT mice. **m** Representative lateral X-ray images of WT and LONP1 mKO mice at indicated age. $n = 3$–9 mice per group. Values represent mean ± SEM; for (**b**) and (**e**), *$P < 0.05$, **$P < 0.01$, ***$P < 0.001$ versus corresponding WT controls, ‡$P < 0.05$, ‡‡$P < 0.01$, ‡‡‡$P < 0.001$ versus Basal, determined by one-way ANOVA coupled to a Fisher's LSD post-hoc test; for (**c**) and (**f-k**), *$P < 0.05$, **$P < 0.01$, ***$P < 0.001$ versus corresponding WT controls determined by two-tailed unpaired Student's $t$ test. Source data are provided as a Source Data file.

subsarcolemmal (SS) compartments of the soleus muscles of LONP1 mKO and WT control mice (Fig. 3a, b and Supplementary Fig. 3a). The quantitative EM analysis revealed that the relative mitochondrial area per muscle fiber area and average mitochondrial size were not different in LONP1 mKO muscle compared with WT controls (Supplementary Fig. 3a,b). Similarly, the mitochondrial DNA content and Tim23 protein levels, as well as PGC-1α transcripts were not different in LONP1 mKO muscle relative to WT controls (Fig. 3c and Supplementary Fig. 3c, d). However, this EM analysis of LONP1 mKO muscles revealed remarkable heterogeneity in mitochondrial structure, with many disturbed mitochondrial ultrastructure that has abnormal cristae organization already at 2 weeks of age that persisted at 6 weeks of age in LONP1 mKO mice (Fig. 3a, b). Mitochondria in WT mice had the regular, accordion-like folds of cristae, whereas both IMF and SS mitochondria in LONP1 mKO muscle showed many abnormal cristae structure: cristae were dilated, frequently lost, or appeared vesiculated (Fig. 3a, b). Furthermore, many mitochondria in LONP1 mKO muscle contained electro-dense aggregates (Fig. 3a, b), which had also been shown in LONP1 knockdown studies in cultured cells[29]. These data indicate that loss of LONP1 alters skeletal muscle mitochondrial function without affecting mitochondrial biogenesis.

Respiration rates were determined on mitochondria isolated from muscle of LONP1 mKO and WT control mice using pyruvate or succinate as substrates. In line with the dramatic structural derangements, both pyruvate/succinate-driven and maximal ADP-stimulated mitochondrial respiration rates were markedly reduced in LONP1 mKO muscles compared to the WT controls (Fig. 3d, e). We also analyzed the mitochondrial respiratory chain complexes in LONP1 mKO muscles using blue-native gel electrophoresis (BN-PAGE). Loss of LONP1 in skeletal muscle led to a mild-to-moderate decrease in the amount of fully assembled complexes I, III, and IV (Supplementary Fig. 4a). Similarly, we also detected a mild decrease in in-gel activities of complexes I and IV in LONP1 mKO muscles relative to WT controls (Supplementary Fig. 4b). Consistent with impaired mitochondrial respiration capacity, ATP levels were significantly lower in LONP1 mKO muscles compared to WT controls (Fig. 3f). This was accompanied by activation of AMPK in LONP1 mKO mice (Supplementary Fig. 3e). Together, these data demonstrate that LONP1 is an essential regulator in safeguarding mitochondrial function in skeletal muscle.

**Muscle LONP1 ablation does not affect protein synthesis and ubiquitin-mediated protein degradation but activates autophagy.** We believe that impaired muscle mitochondrial function in 2-week-old LONP1 mKO mice acts as a very early signal that precedes other changes observed in LONP1-deficient muscle. The control of skeletal muscle mass is determined by the dynamic balance between anabolic and catabolic processes[11,14]. To dissect the mechanism by which LONP1 abrogation affected skeletal muscle mass maintenance, we first analyzed by quantitative RT-PCR profiling the mRNA expression of muscle atrophy-related genes in 2-week-old mice. No increases were observed in the mRNA abundance of *MAFbx*, *MuRF1*, *FbxO31*, *Itch*, *SMART*, and *MUSA1* in GC muscles from LONP1 mKO mice (Fig. 4a). In fact, the gene expression of the ubiquitin degradation system was actually reduced in muscle from LONP1 mKO mice at 6 weeks, as well as at 16 weeks of age (Fig. 4a and Supplementary Fig. 5b). Foxo3, a regulator of skeletal muscle mass, was shown to promote both ubiquitin-proteasome and autophagy-lysosome systems[30]. We did not detect a change in the phosphorylation of Foxo3 proteins in muscles from LONP1 mKO mice (Fig. 4b). Similarly, the content of ubiquitinated proteins in muscle homogenates

from WT and LONP1 mKO mice was not different at 2 or 16 weeks of age (Fig. 4c and Supplementary Fig. 5a, c). We also monitored protein synthesis rate in vivo by using the SUnSET technique, and we found protein synthesis was not different in LONP1 mKO mice compared with that in WT controls (Fig. 4d and Supplementary Fig. 5d). Furthermore, we confirmed that the AKT/mTOR axis, signaling in controlling muscle anabolic process, was not changed between two genotypes (Fig. 4e). Together, these data indicate that LONP1 ablation does not affect muscle protein synthesis and ubiquitin-mediated protein degradation.

The autophagy-lysosome pathway is another major route for protein and organelle clearance in skeletal muscle. We thus turned to monitor the status of muscle autophagy. We bred LONP1 mKO mice with a reporter mouse containing the fluorescent reporter mt-Keima to assess mitochondrial autophagy in skeletal muscle lacking LONP1. We found a significant increase in mitochondrial autophagy already at 2 weeks of age that became more dramatic at 6 weeks of age in LONP1 mKO mice (Fig. 4f, g and Supplementary Fig. 5e). The skeletal muscle of mt-Keima Tg mice had few red fluorescence signals confined to small, round punctate structures (Fig. 4f, g and Supplementary Fig. 5e), indicating low levels of basal mitochondrial autophagy. This red mt-Keima fluorescence signal was, however, dramatically induced in LONP1 mKO/mt-Keima muscle (Fig. 4f, g and Supplementary Fig. 5e), suggesting that LONP1 deficiency triggers a mitochondrial autophagy program. To further confirm the induced autophagy program in LONP1 mKO muscle, we performed autophagy flux analysis by treating mice with colchicine, an established inhibitor of lysosomal degradation. As shown in Fig. 4h, LC3-mediated autophagy flux was significantly induced in WV muscle mitochondrial fraction isolated from LONP1 mKO mice. Moreover, colchicine treatment caused a greater accumulation of LC3-II and P62 protein in whole muscle homogenates from LONP1 mKO mice compared with that in WT controls (Fig. 4i), indicating an enhanced general autophagy flux in LONP1 mKO muscles. In addition, the effects of autophagy inhibition using chloroquine (CQ) were assessed in WT and LONP1 mKO mice. Analysis of fiber size distribution revealed that CQ treatment led to a reduced number of small fibers and an increased number of larger fibers in LONP1 mKO mice (Supplementary Fig. 5f, g). Transcription factor EB (TFEB) is a regulator of lysosomal biogenesis. We did not detect a change in TFEB nuclear translocation in muscles from LONP1 mKO mice (Supplementary Fig. 5i). In addition, the expression of LAMP1 protein was not increased in LONP1 mKO muscles versus WT controls (Supplementary Fig. 5h). These data suggest that LONP1 ablation might not activate lysosomal biogenesis in skeletal muscles. Together, these results demonstrate that loss of LONP1 induces the activation of autophagy-lysosome degradation program of muscle loss.

**Acute deletion of LONP1 in mature muscle leads to autophagy activation and causes muscle loss and weakness.** To test whether depletion of LONP1 in mature muscle could also activate autophagy and cause reduced muscle fiber size and strength, we performed adeno-associated virus (AAV) Cre-mediated LONP1 deletion in skeletal muscle of *Lonp1*[f/f] mice, and we conducted terminal studies 8 weeks after the injection of AAV-Cre in 4-week-old mice. Compared to *Lonp1*[f/f] muscle injected with control AAV-GFP viruses, AAV-Cre-mediated deletion of LONP1 in skeletal muscle resulted in lower levels of LONP1 but not CLPP protein (Fig. 5a). Consistent with the results from LONP1 mKO mice, AAV-based LONP1 ablation in *Lonp1*[f/f] muscle also caused muscle weight loss and muscle weakness (Fig. 5b–d). Indicative of activated autophagy, the number of red mt-Keima puncta was

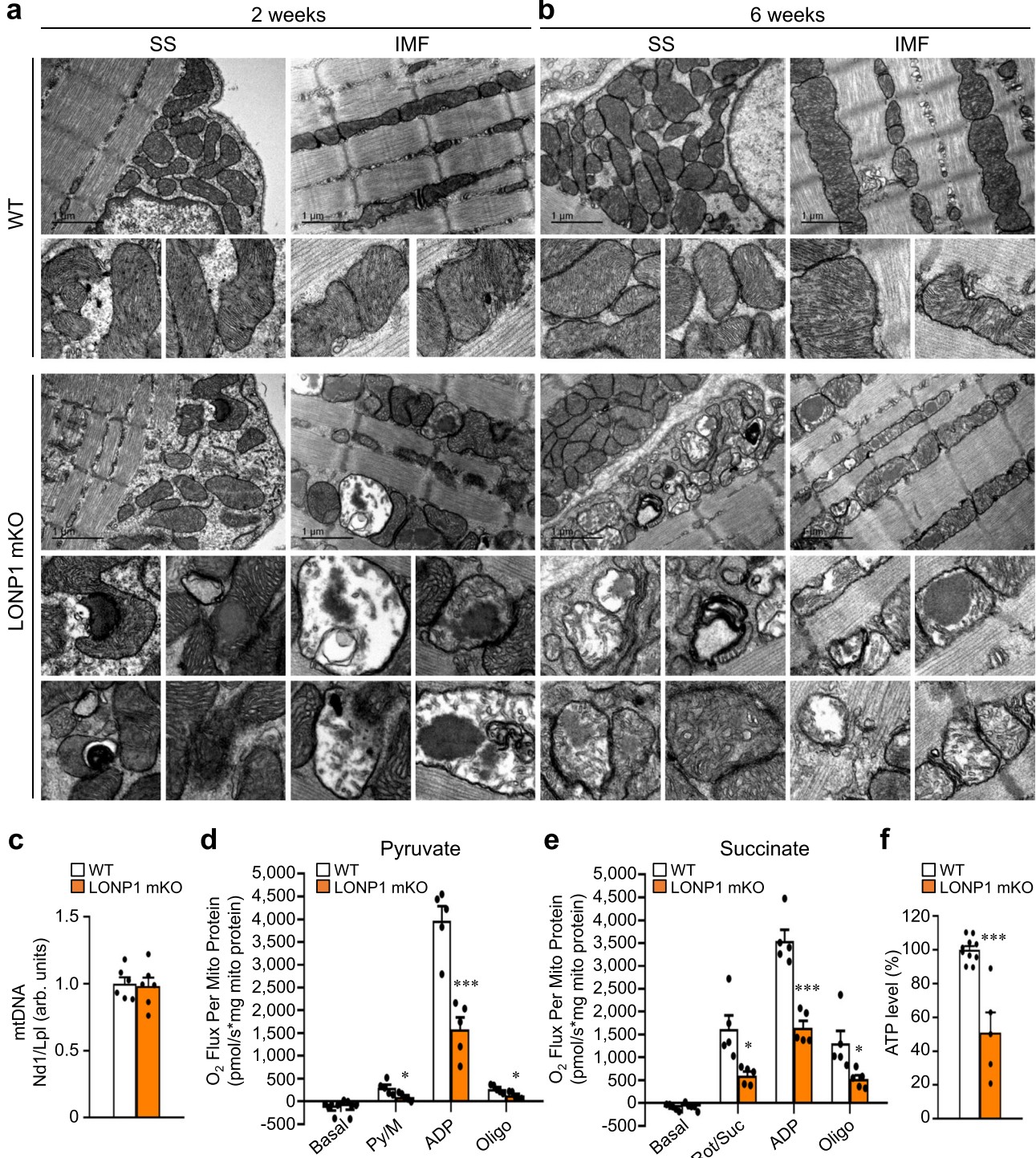

**Fig. 3 Loss of LONP1 leads to severe mitochondrial structural and functional abnormalities in skeletal muscle. a, b** Representative electron micrographs of soleus muscle showing subsarcolemmal (SS) and intermyofibrillar (IMF) mitochondria in sections from WT and LONP1 mKO mice at the age of 2 weeks (**a**) and at 6 weeks (**b**). The scale bar represents 1 μm. $n = 4$–6 mice per group. Note, dilated, frequently lost, or appeared vesiculated cristae and electro-dense aggregates were observed in LONP1 mKO mitochondria. No abnormal mitochondrial ultrastructures were observed in the WT controls. **c** Results of quantitative PCR to determine mitochondrial DNA levels in GC muscle of indicated mice at 6 weeks of age using primers for NADH dehydrogenase (Nd1, mitochondria-encoded) and lipoprotein lipase (Lpl, nuclear-encoded). Nd1 levels were normalized to Lpl DNA content and expressed relative to WT (=1.0) muscle. $n = 6$ mice per group. **d, e** Mitochondrial respiration rates were determined on mitochondria isolated from WV muscles of indicated mice using pyruvate (**d**) or succinate (**e**) as substrates. Pyruvate/malate (Py/M) or succinate/rotenone (Suc/Rot)-stimulated, ADP-dependent respiration, and oligomycin (oligo)-induced are shown. $n = 5$ mice per group. **f** ATP levels were detected and normalized to the percentage of WT muscles. $n = 5$ mice per group. Values represent mean ± SEM; for (**c**–**f**), ***$P < 0.001$ versus corresponding WT controls determined by two-tailed unpaired Student's $t$ test. Source data are provided as a Source Data file.

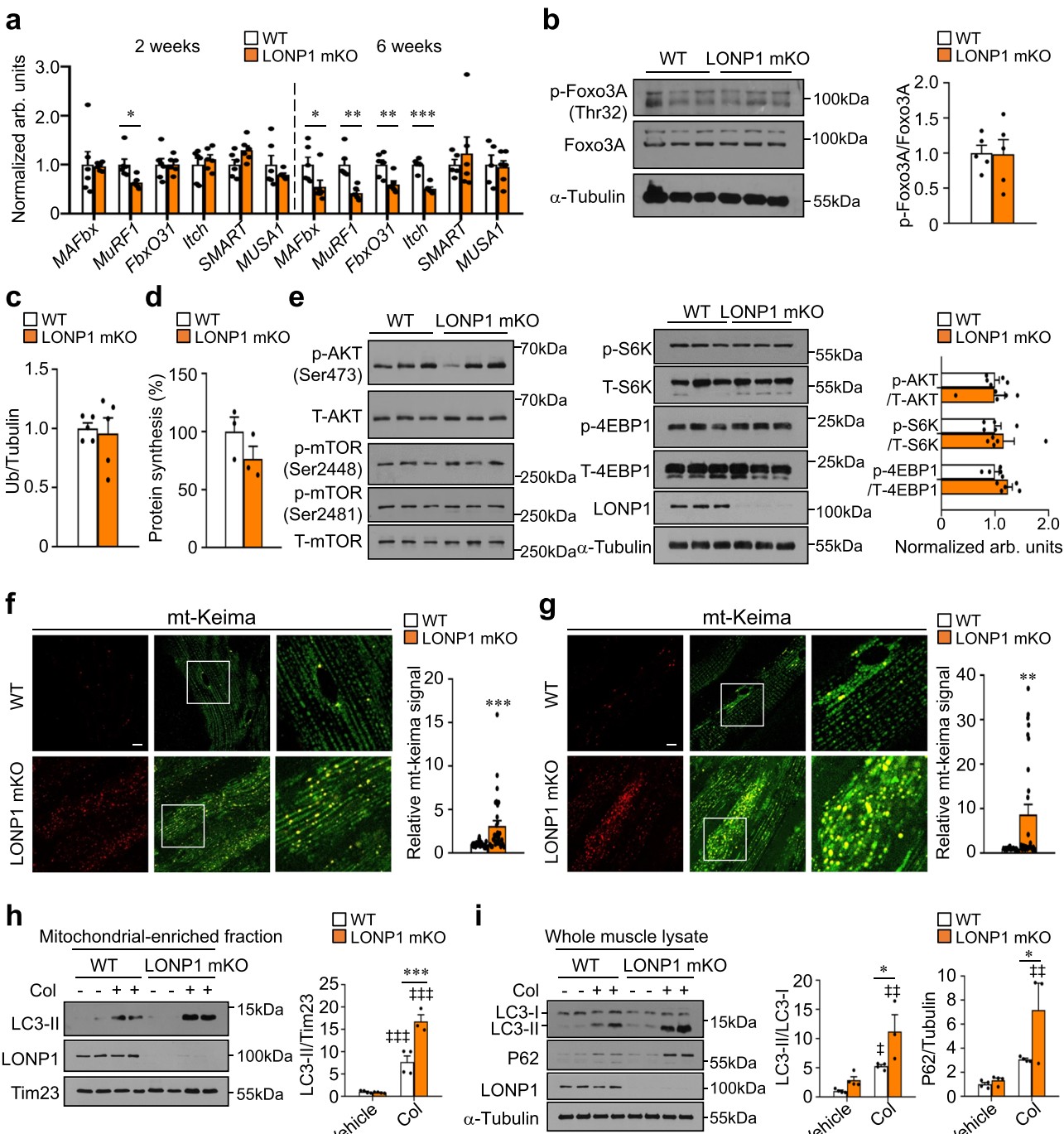

**Fig. 4 Muscle LONP1 deficiency does not affect protein synthesis and ubiquitin-mediated protein degradation but activates autophagy. a** Expression of atrophy-related genes (RT-qPCR) in GC muscle from indicated mice at the age of 2 weeks and at 6 weeks. n = 5–6 mice per group. **b** (Left) Western blot analysis of WT and LONP1 mKO WV muscles using indicated antibodies. (Right) Quantification of the p-Foxo3A/Foxo3A signal ratios. *n* = 5 mice per group. **c** Quantification of the ubiquitinated proteins of muscle total protein extracts from 2-week-old indicated mice. *n* = 5 mice per group. **d** Quantification of puromycin incorporation in muscles from indicated mice at the age of 2 weeks. *n* = 3 mice per group. **e** (Left) Western blot analysis of WV muscles from 2-week-old mice using indicated antibodies. (Right) Quantification of the p-AKT/AKT, p-S6K/S6K and p-4EBP1/4EBP1 signal ratios. *n* = 5 mice per group. **f, g** (Left) Confocal images of TA (**f**) and EDL (**g**) muscles in 6-week-old mt-Keima-TG mice and LONP1 mKO/mt-Keima mice. The scale bar represents 10 μm. 458-nm laser, green, 561-nm laser, red. (Right) Quantification of the red/green mt-Keima signal ratios. *n* = 3 mice per group. **h, i** Autophagy flux is increased in LONP1 mKO muscles. **h** (Left) Western blot analysis of mitochondrial-associated LC3-II in WV muscle from indicated mice. Mice were treated for 24 h with colchicine (Col) as indicated. (Right) Quantification of the mitochondrial-associated LC3-II/Tim23 signal ratios. *n* = 3–4 mice per group. **i** (Left) Western blot analysis of WV muscle total protein extracts prepared from indicated mice treated for 24 h with or without Col. (Right) Quantification of the LC3-II/LC3-I and P62/Tubulin signal ratios. *n* = 3–4 mice per group. Values represent mean ± SEM; for (**a–g**), *P < 0.05, **P < 0.01, ***P < 0.001 versus corresponding controls determined by two-tailed unpaired Student's *t* test; for (**h**) and (**i**), *P < 0.05, ***P < 0.001 versus corresponding controls, ‡‡P < 0.01, ‡‡‡P < 0.001 versus vehicle controls, determined by one-way ANOVA coupled to a Fisher's LSD post-hoc test; Source data are provided as a Source Data file.

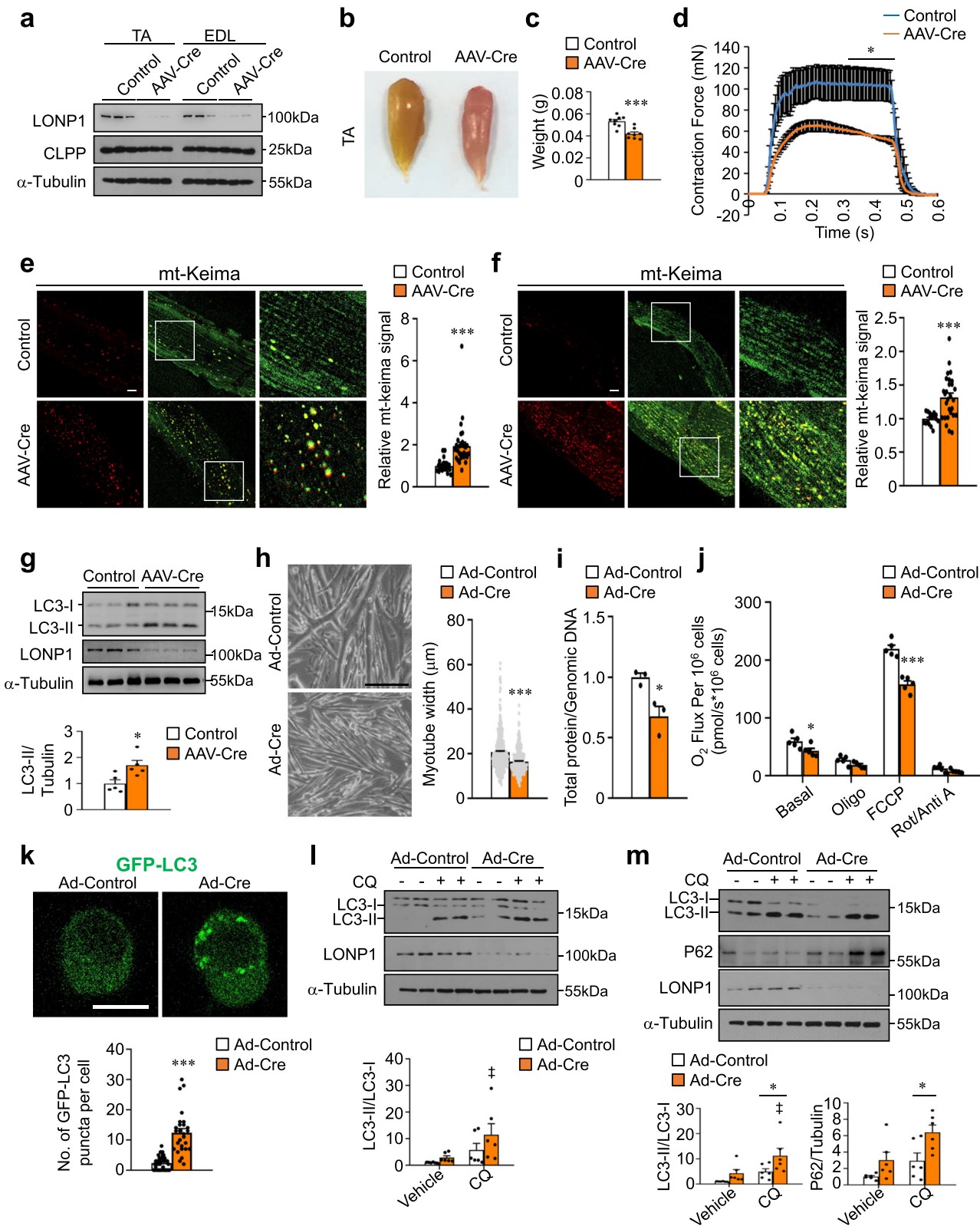

significantly increased in TA and EDL muscle from *Lonp1*[f/f]/mt-Keima injected with AAV-Cre compared to control viruses (Fig. 5e, f). The activation of autophagy was further confirmed by the increased LC3-II levels in whole muscle homogenates lacking LONP1 (Fig. 5g). Therefore, loss of LONP1 in mature muscles also causes autophagy activation and muscle loss. The fact that two mouse models lacking LONP1 in skeletal muscle show similar reduced muscle mass phenotype strongly suggests that LONP1 is a regulator for preserving skeletal muscle mass and strength.

In an additional step, an LONP1 loss-of-function study was also conducted in vitro primary skeletal myocyte culture system. We transduced adenovirus expressing Cre or control into *Lonp1*[f/f] myoblasts. Cells were then induced to differentiate

**Fig. 5 Acute deletion of LONP1 in mature muscle leads to autophagy activation and causes muscle loss and weakness. a–g** Mice were sacrificed 8 weeks after the injection of AAV-Cre or AAV-Control in 4-week-old *Lonp1*f/f mice. **a** Western blot analysis of muscles for *Lonp1*f/f mice transduced with indicated AAV. $n = 8$ mice per group. **b** TA muscles from indicated mice. **c** TA muscle weight from indicated mice. $n = 8$ mice per group. **d** Isometric tetanic contractions measurement. $n = 3$ mice per group. **e, f** (Left) Confocal images of TA (**e**) and EDL (**f**) muscles for *Lonp1*f/f/mt-Keima mice transduced with indicated AAV. Scale bar: 10 μm. (Right) Quantification of the red/green mt-Keima signal ratios. $n = 3$ mice per group. **g** (Top) Western blot analysis of TA muscles from indicated mice. (Bottom) Quantification of the LC3-II/Tubulin signal ratios. $n = 5$ mice per group. **h–i** Primary myoblasts isolated from *Lonp1*f/f mice were infected with an adenovirus overexpressing Cre or control virus, followed by differentiation into myotubes. **h** (Left) Images of myotubes. Scale bar: 200 μm. (Right) Measurements of myotube diameter. $n = 376–418$ myotubes per condition. **i** Total cellular protein content relative to genomic DNA in myotubes. $n = 3$. **j** Oxygen consumption rates (OCR) in myotubes. $n = 5$. **k** (Top) Confocal images of primary myoblasts expressing GFP-LC3. Scale bar: 10 μm. (Bottom) Quantification of GFP puncta per cell. $n = 28–41$ cells. **l, m** Autophagy flux is increased in LONP1 KO myocytes. (Top) Western blot analysis of myoblasts (**l**) and myotubes (**m**) subjected to adenovirus-based overexpression of Cre compared with control. Myocytes were treated for 4 h with or without chloroquine (CQ) as indicated. (Bottom) Quantification of the LC3-II/LC3-I and P62/Tubulin signal ratios. $n = 3$. Values represent mean ± SEM; for (**c–k**), *$P < 0.05$, ***$P < 0.001$ versus corresponding controls determined by two-tailed unpaired Student's *t* test; for (**l**) and (**m**), *$P < 0.05$ versus corresponding controls, ‡$P < 0.05$ versus vehicle controls, determined by one-way ANOVA coupled to a Fisher's LSD post-hoc test; Source data are provided as a Source Data file.

into myotubes. Myotubes expressing Cre resulted in a decrease in myotube diameter compared to those expressing control vector (Fig. 5h), with a 45% reduction in the ratio of total protein to genomic DNA (Fig. 5i). Notably, we observed no differences in myotube differentiation between two groups (Fig. 5h). As expected, in vitro deletion of LONP1 in myotubes resulted in impaired mitochondrial respiration as observed in LONP1 mKO muscle (Fig. 5j). A decrease in the amount of fully assembled complexes IV was also observed in LONP1 KO myotubes (Supplementary Fig. 6e). There were no changes in ubiquitin-mediated protein degradation, protein synthesis rates, and AKT/mTOR signaling in WT and LONP1 KO myotubes (Supplementary Fig. 6a–d). LONP1 deficiency induced early autophagy activation (GFP-LC3 puncta) in myoblasts (Fig. 5k). Moreover, analysis of autophagy flux in both LONP1 KO myoblasts and myotubes showed similar autophagy activation to those in vivo in LONP1 mKO muscle (Fig. 5l, m). Thus, the in vitro manipulation of LONP1 in muscle cells confirmed most of the data obtained by the LONP1 mKO muscle. Taken together, these results demonstrate that loss of LONP1 activates the autophagy degradation program of muscle loss both in vivo and in vitro.

We next examined whether the LONP1-dependent regulation of the autophagy program is operational during denervation-induced muscle loss. Consistent with reduced LONP1 expression, we found that the red mt-Keima fluorescence signal was significantly induced in EDL muscles of mt-Keima Tg mice at 5 days following denervation (Supplementary Fig. 7a). In addition, increased conversion of LC3-I to LC3-II and decreased mitochondrial respiration capacity were detected in muscles following denervation (Supplementary Fig. 7b, c). Notably, increased expression of ubiquitin degradation genes (e.g. *MAFbx*, *MuRF1*) were also seen at early stages of denervation (Supplementary Fig. 7d), which is consistent with previous reports[31,32]. These data indicate that activation of the LONP1-dependent autophagy pathway, along with the ubiquitin degradation system, accompanies the process of disuse-induced skeletal muscle loss. We also investigated whether other autophagy signaling such as mTOR-dependent pathway is involved in LONP1-mediated regulation of muscle function. We explored the impact of rapamycin treatment in AAV-Cre-mediated LONP1-knockout model. Rapamycin was administered intraperitoneally (8 mg/kg/day) for 4 weeks. Fiber area frequency distribution revealed that rapamycin treatment led to an increase in the percentage of small fibers (a leftward shift) in both LONP1-knockout and control mice (Supplementary Fig. 8a–c). These results suggest an mTORC1-independent regulation of muscle function by LONP1.

**LONP1 controls mitochondrial proteostasis that regulates muscle cell autophagy.** How does loss of LONP1-mediated mitochondrial protein quality control affect skeletal muscle function? We addressed the potential pathogenetic involvement of mitochondrial proteostasis. We bred LONP1 mKO mice with MitoTimer reporter mice to monitor the overall protein turnover status of the mitochondrial reticulum in LONP1 mKO skeletal muscles. We found that the red:green ratio of the MitoTimer was significantly shifted towards red fluorescence in both TA and EDL muscles of LONP1 mKO mice (Fig. 6a, b), indicating an impaired mitochondrial protein turnover in LONP1 mKO muscles. We further analyzed the mitochondrial protein changes in the presence and absence of LONP1. Proteomes of isolated muscle mitochondria were analyzed by isobaric tags for relative and absolute quantification (iTRAQ) mass spectrometry. Based on MitoCarta2.0[33], we were able to identify 618 mitochondrial proteins, of which 81 (13%) displayed 1.2 fold changes in abundance upon LONP1 deletion compared with WT mitochondria (Fig. 6c). Consistent with the role of LONP1 in degrading mitochondrial protein, we detected more upregulated than downregulated proteins (55 and 26) upon LONP1 depletion (Supplementary Fig. 9a), and the heatmap analysis of the duplicate samples showed the similarity between replicates (Fig. 6d). Furthermore, proteins known to localize to the mitochondrial matrix or mitochondrial inner membrane involved in respiration, translation, and protein quality control were found to be regulated in LONP1 deletion mitochondria (Fig. 6d).

We then proceeded to analyze possible LONP1 substrates involved in the regulation of autophagy. Interestingly, PARK7 protein, whose expression was upregulated in LONP1 mKO muscle mitochondria, was recently shown to regulate autophagy and degraded by LONP1[34–37]. We sought to determine whether PARK7 protein is involved in LONP1-mediated regulation of muscle cell autophagy. We found that PARK7 protein levels were markedly increased (5-fold) in LONP1 KO myotubes (Fig. 6e). However, *Park7* mRNA abundance was comparable in LONP1 KO myotubes versus WT controls (Fig. 6f), pointing to a post-transcriptional regulation of PARK7 protein. Notably, PINK1 protein levels remained unchanged in LONP1 KO myotubes compared to WT controls (Fig. 6e). Moreover, cycloheximide chase experiments revealed an increased stability of PARK7 protein in LONP1 KO myotubes (Fig. 6g), suggesting that PARK7 is likely LONP1 substrate. We also conducted co-immunoprecipitation (co-IP) studies to determine whether LONP1 directly interacts with PARK7. HEK293T cells were cotransfected with expression vectors for Flag-PARK7 and LONP1. Anti-Flag was found to co-immunoprecipitation PARK7 and LONP1 (Fig. 6h). Together, these data suggest that PARK7 is

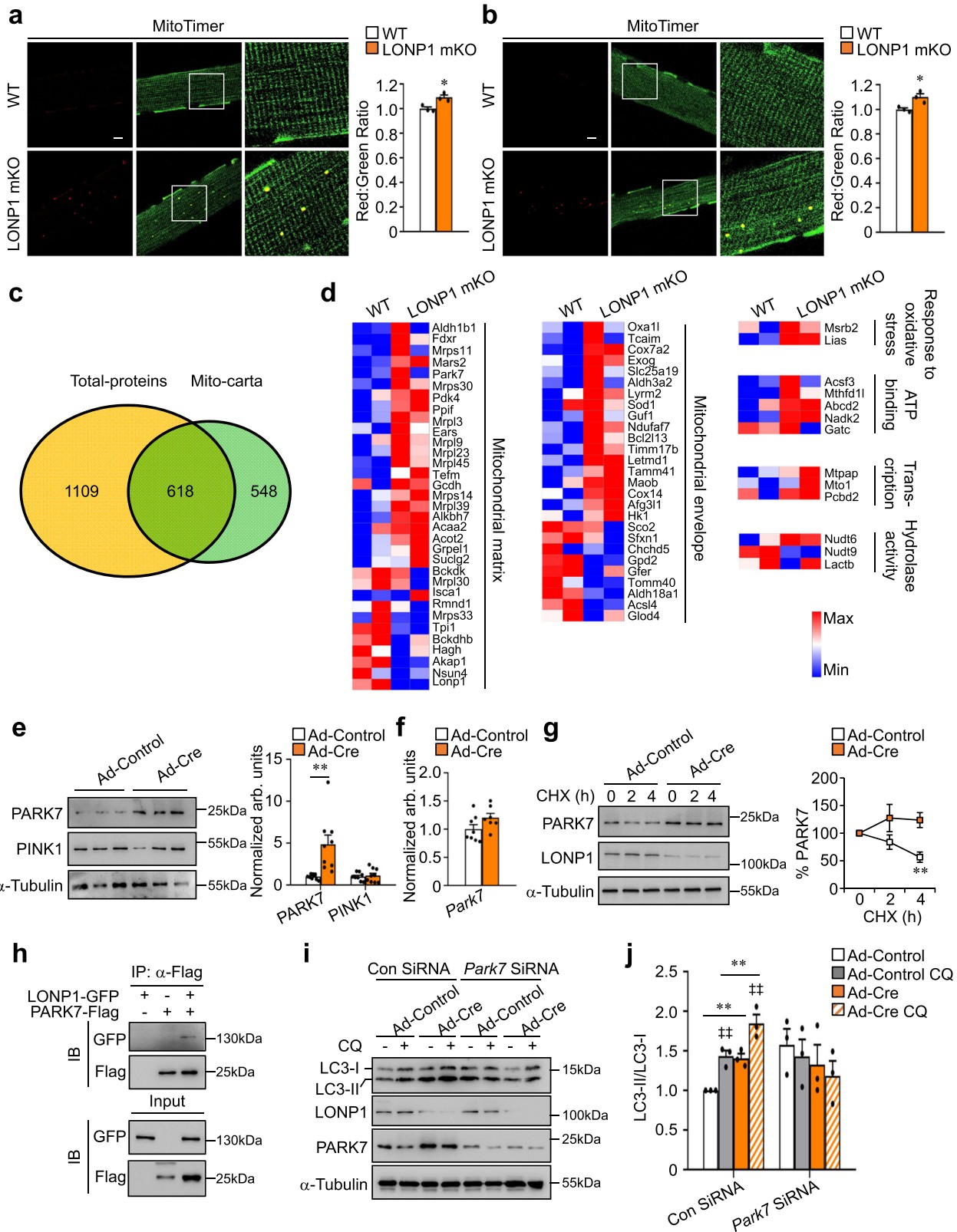

a LONP1 substrate. We then tested whether PARK7 linked LONP1 deficiency to muscle cell autophagy. Loss-of-function experiments in primary myotubes showed that siRNA knockdown of *Park7* reduced autophagy flux in LONP1 KO myotubes (Fig. 6i, j). Thus, PARK7 mediates autophagy activation in LONP1-deficient cells, thereby linking LONP1-dependent mitochondrial proteostasis to muscle cell autophagy.

**LONP1 preserves muscle function through maintaining mitochondrial proteostasis**. We then hypothesized that mitochondria overloaded by unfolded proteins beyond the LONP1 capacity might also lead to similar muscle effects. To test this hypothesis, we generated transgenic mice overexpressing a deletion mutant of ornithine transcarbamylase (ΔOTC), a known protein degraded by LONP1 and an established model for studying mitochondrial

**Fig. 6 LONP1 controls mitochondrial proteostasis that regulates muscle cell autophagy. a, b** (Left) Confocal images of TA (**a**) and EDL (**b**) muscles from LONP1 mKO/MitoTimer and *Lonp1*f/f/MitoTimer mice. The scale bar represents 10 µm. (Right) Quantification of MitoTimer Red:Green ratio. n = 3 mice per group. **c, d** Proteomes of crude mitochondria isolated from 2-week-old WT and LONP1 mKO muscles were analyzed by isobaric tags for relative and absolute quantification (iTRAQ) mass spectrometry. n = 2 independent samples per group. **c** Venn diagram representing the proteomics results compared with MitoCarta2.0 database. **d** Heat-maps analysis of mass spectrometry proteomics data. Individual proteins are shown to be regulated in the LONP1 mKO muscles as denoted by the color scheme. **e** (Left) Western blot analysis of myotubes subjected to adenovirus-based overexpression of Cre compared with control. (Right) Quantification of the PARK7/Tubulin and PINK1/Tubulin signal ratios. n = 3. **f** RT-qPCR analysis of *Park7* in myotubes. n = 3. **g** (Left) Cycloheximide (CHX) treatment of myotubes for the indicated time. (Right) Quantification of the PARK7 protein levels is shown. n = 3. **h** Co-IP experiments were performed by cotransfecting LONP1-GFP and PARK7-Flag in HEK293 cells as indicated at the top. Antibody against the Flag epitope was used for co-IP. The extracts (Input) from the HEK293 cells and proteins from the IP were analyzed by immunoblotting (IB). Representative results for co-IP are shown. n = 3. **i** Western blot analysis of LC3-II/LC3-I ratios in myotubes transfected with control siRNA or *Park7* siRNA and subjected to adenovirus-based overexpression of Cre compared with control. Myotubes were treated for 4 h with or without chloroquine (CQ) as indicated. **j** Quantification of the LC3-II/LC3-I signal ratios in (**i**). n = 3. Values represent mean ± SEM; for (**a**), (**b**), (**e**), (**f**) and (**g**), *P < 0.05, **P < 0.01 versus corresponding controls determined by two-tailed unpaired Student's t test; for (**j**), **P < 0.01 versus corresponding controls, ‡‡P < 0.01 versus vehicle controls, determined by one-way ANOVA coupled to a Fisher's LSD post-hoc test; Source data are provided as a Source Data file.

proteostasis imbalance[38–40], specifically in skeletal muscle (MCK-ΔOTC). MCK-ΔOTC mice showed ΔOTC overexpression in skeletal muscle mitochondrial fraction compared with that in NTG controls (Fig. 7a). Increased expression of LONP1 protein was detected upon overexpression of ΔOTC in skeletal muscles (Supplementary Fig. 9b), indicating mitochondrial proteostasis stress in MCK-ΔOTC muscles. Respiration rates were next determined on mitochondria isolated from WV muscle of MCK-ΔOTC and NTG control mice using pyruvate or succinate as substrates. Both pyruvate/succinate-driven and maximal ADP-stimulated mitochondrial respiration rates were markedly reduced in MCK-ΔOTC muscles compared to the NTG controls (Fig. 7b, c). Together, these data suggest that the accumulation of misfolded proteins in mitochondria is a physiological trigger of muscle mitochondrial dysfunction.

Next, we further determined whether ΔOTC overexpression could activate the autophagy degradation program of muscle loss. As shown in Fig. 7d, skeletal muscle-specific ΔOTC-overexpressing mice showed reduced muscle size (Fig. 7d). GC and TA muscle weights were significantly lower in MCK-ΔOTC mice compared to NTG control littermates (Fig. 7e). Examination of myofiber cross-sectional area by histochemical staining revealed a marked decrease in myofiber size compared with that in NTG controls (Fig. 7f). This was accompanied by a significant reduction of muscle force in MCK-ΔOTC muscles compared to NTG controls (Fig. 7g). Thus, overexpression of ΔOTC in skeletal muscle is sufficient to cause reduced muscle mass and strength. In agreement with no induction of ubiquitin-mediated protein degradation or suppression of protein synthesis in LONP1 mKO muscles, we confirmed that the expression of atrophy-related genes, Foxo3 protein, and ubiquitinated proteins content, as well as the AKT/mTOR signaling were not different in MCK-ΔOTC muscles compared to NTG controls (Fig. 7h, i and Supplementary Fig. 9c, d). To further test the effects of ΔOTC overexpression on muscle catabolism, we determined whether ΔOTC overexpression can trigger the autophagy degradation program in skeletal muscle. As shown in Fig. 7j, the conversion of LC3-I to LC3-II was increased in whole muscle homogenates overexpressing mitochondrial ΔOTC (Fig. 7j). The activation of muscle autophagy program was also confirmed by the increased red mito-Keima puncta in MCK-ΔOTC/mito-Keima muscles compared to mt-Keima Tg mice (Fig. 7k, l). Together, these data demonstrate that muscle-specific overexpressing mitochondrial-retained ΔOTC protein is sufficient to induce mitochondrial dysfunction, autophagy activation, and cause muscle loss and weakness.

We next investigated whether overexpression of LONP1 could impact the muscle phenotype in MCK-ΔOTC mice. Consistent with LONP1 degrading ΔOTC[39,40], overexpression of LONP1 using AAV resulted in a significant reduction of ΔOTC protein in MCK-ΔOTC muscles (Fig. 8a, b). This was accompanied by reduced amount of LC3-II that was increased in MCK-ΔOTC muscles (Fig. 8c). In agreement with the role of LONP1 in preserving muscle mitochondrial function, increased expression of LONP1 resulted in significantly increased mitochondrial respiration capacity in MCK-ΔOTC muscles (Fig. 8d). Quantification of muscle fiber size distribution revealed that overexpression of LONP1 also led to a shift toward larger myofibers in MCK-ΔOTC mice (Fig. 8e, f). Together, these results further suggest that LONP1 plays a role in preserving muscle function through maintaining mitochondrial proteostasis.

## Discussion

Mitochondria are fundamental to muscle functions. Physical inactivity-associated muscle loss and weakness is a major health problem and is related to a variety of chronic human diseases. Preventing or reversing muscle loss is of great importance to improve the quality of life and reduce mortality, yet the mechanisms involved in disuse-induced muscle loss and weakness are largely unknown. In this study, we described an essential role of mitochondrial protease LONP1 in the regulation of mitochondrial function and skeletal muscle mass in response to muscle disuse (Fig. 8g). We discovered that muscle LONP1 protein is sensitive to muscle disuse in mice and humans. Deletion of LONP1 specifically in skeletal muscle resulted in impaired mitochondrial protein turnover, leading to mitochondrial dysfunction and significant reduced muscle mass and strength. Importantly, our data demonstrate that aberrant accumulation of mitochondrial-retained protein in muscle upon loss of LONP1 induces the activation of autophagy-lysosome degradation program of muscle loss. Moreover, overexpressing a mitochondrial-retained ΔOTC protein in skeletal muscle is sufficient to induce mitochondrial dysfunction, autophagy activation, and causes muscle atrophy and weakness. Thus, these findings reveal a role of LONP1-dependent mitochondrial protein quality-control in safeguarding mitochondrial function and preserving skeletal muscle mass and strength, and unravel a functional link between mitochondrial protein quality control and muscle mass maintenance during muscle disuse. This LONP1 mechanism likely acts as a mitochondrial sensor of cellular stress under an array of diverse physiological and pathophysiological circumstances.

Mitochondrial proteases have been considered as the first-line mitochondrial quality-control mechanism in mammalian cells. The importance of the regulated mitochondrial protein degradation is highlighted by a number of human genetic diseases that are associated with mutations in mitochondrial proteases[1,2,4,5],

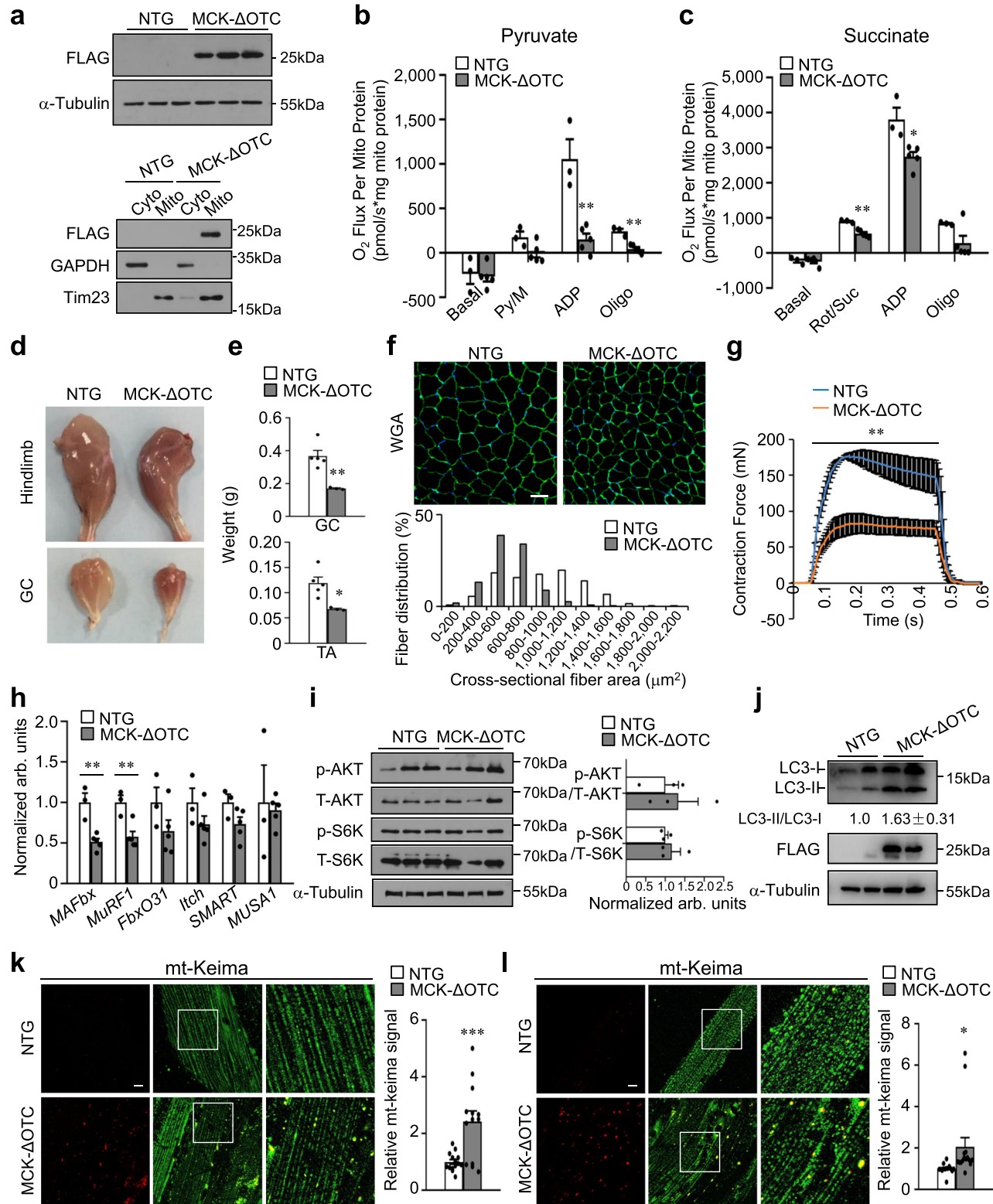

but our understanding of the in vivo physiological relevance and mechanisms of stress-induced dysregulated mitochondrial protein quality remains largely convoluted. Both ATP-dependent proteases LONP1 and CLPP are stationed in the mitochondrial matrix and regulate mitochondrial protein degradation. In this study, we clearly showed that LONP1 is the dominant protease responsible for maintaining mitochondrial function and skeletal muscle mass. Genetic ablation of LONP1 in skeletal muscle

resulted in severe mitochondrial dysfunction and caused reduced muscle fiber size and strength, leading to a precocious aging phenotype. Importantly, AAV-Cre-mediated acute deletion of LONP1 in mature muscle also results in muscle loss and muscle weakness. Moreover, most of these findings are recapitulated in skeletal myotubes, thus, are cell-autonomous. It is possible that LONP1 could compensate for the loss of CLPP in skeletal muscle, given that muscle-specific deletion of CLPP using muscle creatine

**Fig. 7 Overexpression of mitochondrial-retained ΔOTC protein in skeletal muscle activates autophagy and causes muscle loss and weakness. a** Western blot analysis of mitochondrial ΔOTC overexpression in WV muscle total protein extracts (Top) or mitochondrial fraction (Bottom) prepared from indicated mice. n = 3 mice per group. **b**, **c** Mitochondrial respiration rates were determined on mitochondria isolated from WV muscles of indicated mice using pyruvate (**b**) or succinate (**c**) as substrates. n = 3–5 mice per group. **d** Picture of hindlimbs and GC muscles from 8-week-old male mice. **e** GC and TA muscle weight from indicated mice. n = 3–5 mice per group. **f** (Top) Representative WGA (green) staining of GC muscle from 8 week old indicated mice. Scale bar represents 50 μm. (Bottom) Cross-sectional areas of GC myofibers were measured by ImageJ. n = 5 mice per group. **g** Isometric tetanic contraction measurements for isolated EDL muscle from indicated mice. n = 3–5 mice per group. **h** Expression of muscle atrophy-related genes (RT-qPCR) in GC muscle from indicated mice. n = 3–5 mice per group. **i**, **j** Representative immunoblot analysis of WV muscle total protein extracts from indicated mice. **i** (Right) Quantification of the p-AKT/AKT and p-S6K/S6K signal ratios were normalized (=1.0) to NTG controls. n = 3 mice per group. **j** Quantification of the LC3-II/LC3-I signal ratios were normalized (=1.0) to NTG controls and presented below the corresponding bands. **k**, **l** (Left) Representative confocal images of TA (**k**) and EDL (**l**) muscles in mt-Keima-TG mice and MCK-ΔOTC/mt-Keima mice. The scale bar represents 10 μm. 458-nm laser, green, 561-nm laser, red. (Right) Quantification of the red/green mt-Keima signal ratios normalized (=1.0) to NTG controls is shown. n = 3 mice per group. Values represent mean ± SEM; for (**b**), (**c**), (**e**), (**g**), (**h**), (**i**), (**k**) and (**l**), *P < 0.05, **P < 0.01, ***P < 0.001 versus corresponding NTG controls determined by two-tailed unpaired Student's t test. Source data are provided as a Source Data file.

kinase promoter-driven Cre does not lead to an overt muscle phenotype compared to WT controls[41]. ΔOTC is a known protein degraded by LONP1 and a well-established model for studying mitochondrial proteostasis imbalance[38–40]. Our findings that muscle-specific overexpressing mitochondrial-retained ΔOTC protein activates autophagy and causes muscle loss revealed an important physiological relevance of imbalanced mitochondrial proteostasis. Importantly, we have demonstrated that the effect of overexpressing ΔOTC in skeletal muscle can be modulated by LONP1, thereby further suggesting that LONP1 plays a role in preserving muscle function through maintaining mitochondrial proteostasis.

Remarkably, our data suggest that muscle LONP1 likely serves as a mitochondrial sensor that is sensitive to changes in physical activity levels, linking maintenance of mitochondrial protein quality and muscle mass to physical activity. We found that loss of LONP1-dependent mitochondrial protein quality-control in skeletal muscle triggers autophagy degradation program, leading to profound impacts on muscle mass and strength. However, a surprising finding of our study was dissociation between mitochondrial function and type I fiber program in LONP1 mKO mice muscle. The observed increase in type I fibers in LONP1 mKO mice is consistent with a previous report that mitochondrial energetic deficiency can trigger compensatory muscle fiber type switching[42]. A number of studies have demonstrated that proper maintenance of mitochondria dynamics, i.e. the balance between fission and fusion, is necessary for optimal muscle mitochondrial function and fitness[20–23,43]. A reduced muscle mass has been reported in mice lacking MFNs. Both transgenic and gene knockout approaches in mice have also been used to demonstrate the important roles of OPA1 in regulating skeletal muscle mass and functions[23,44,45]. Interestingly, only an early reduction in LONP1 protein, but not mitochondrial dynamic proteins, correlates with disuse-induced muscle loss. We have documented the reduced expression of LONP1 protein and impaired mitochondrial protein turnover in disuse muscles from denervation-induced as well as in hindlimb immobilization. In parallel, muscle loss develops in these two conditions. Our data indicate that denervation-induced muscle disuse leads to suppressed expression of LONP1 with increased muscle autophagy and mitochondrial dysfunction, which is consistent with our observations in LONP1 mKO mice. Moreover, loss of LONP1 exacerbated denervation-induced muscle atrophy. We propose that LONP1 may serve as an important modifier in the progression of muscle loss upon muscle disuse and could potentially be targeted to prevent muscle loss. Notably, muscle-specific LONP1-knockout mice display many features of precocious aging including muscle atrophy and kyphosis further suggesting a possible link between impaired muscle mitochondrial protein quality with aging and

sarcopenia. The mechanism whereby LONP1 protein is repressed in disused muscle was not fully delineated in this study, it is possible that the LONP1 protease itself becomes damaged and triggered auto-degradation over time upon stress. Whereas we found that muscle LONP1 deficiency resulted in reduced capacity to utilize fat as a fuel during exercise, which is consistent with marked impaired muscle mitochondrial function in LONP1 mKO mice, it became obvious that aged LONP1 mKO mice are smaller. Notably, we and others have shown that multiple alterations in mitochondrial function such as mitophagy and mitochondrial dynamics can lead to a surprising phenotype characterized by reduced muscle mitochondrial energetics, but remarkably, with resistance to high-fat-diet-induced obesity[21,46]. The seemingly contradictory phenotype is linked to the non-cell-autonomous role of the muscle mitochondrial stress response. Future studies will be necessary to further delineate the LONP1 signaling in regulating muscle mitochondrial stress response and systemic metabolism.

The importance of ubiquitin-proteasome-dependent degradation in muscle atrophy has been well documented[11,14,15]. Interestingly, however, loss of muscle LONP1 activates autophagy degradation program without affecting protein synthesis and ubiquitin-mediated protein degradation. It is possible that LONP1 ablation mainly affects intramitochondrial proteins that are not processed by the ubiquitin-proteasome system. Numerous studies have demonstrated that autophagy is critical for maintaining normal homeostasis of muscle mass in physiological and pathological conditions[30,47–49]. Excessive activation of autophagy aggravates muscle wasting by removing portion of cytoplasm, proteins, and organelles[30,50,51], supporting the idea that activation of autophagy flux in muscle lacking LONP1 leads to muscle loss and weakness. Notably, finely tuned regulation of autophagy is likely necessary for the maintenance of skeletal muscle mass and fitness, as previous studies also suggested that a certain degree of autophagy is necessary to remove damaged proteins and organelles to ensure proper mass and function of skeletal muscle[47,52]. Our data suggest an interesting interconnection between mitochondrial proteolysis and autophagy in skeletal muscle. It turns out that LONP1 is directly involved in mediating muscle cell autophagy. Our mechanistic studies identified PARK7 as a LONP1 substrate, which, when knocked down, reduced autophagy flux in LONP1-deficient muscle cells. These results were intriguing given the recent discoveries that PARK7 protein has been shown to regulate autophagy in multiple cell types[34–36]. However, we cannot exclude the possibility that LONP1 may also act through other mechanisms to regulate autophagy. Interestingly, in contrast to PARK7, our data suggest that LONP1 affects on muscle cell function without

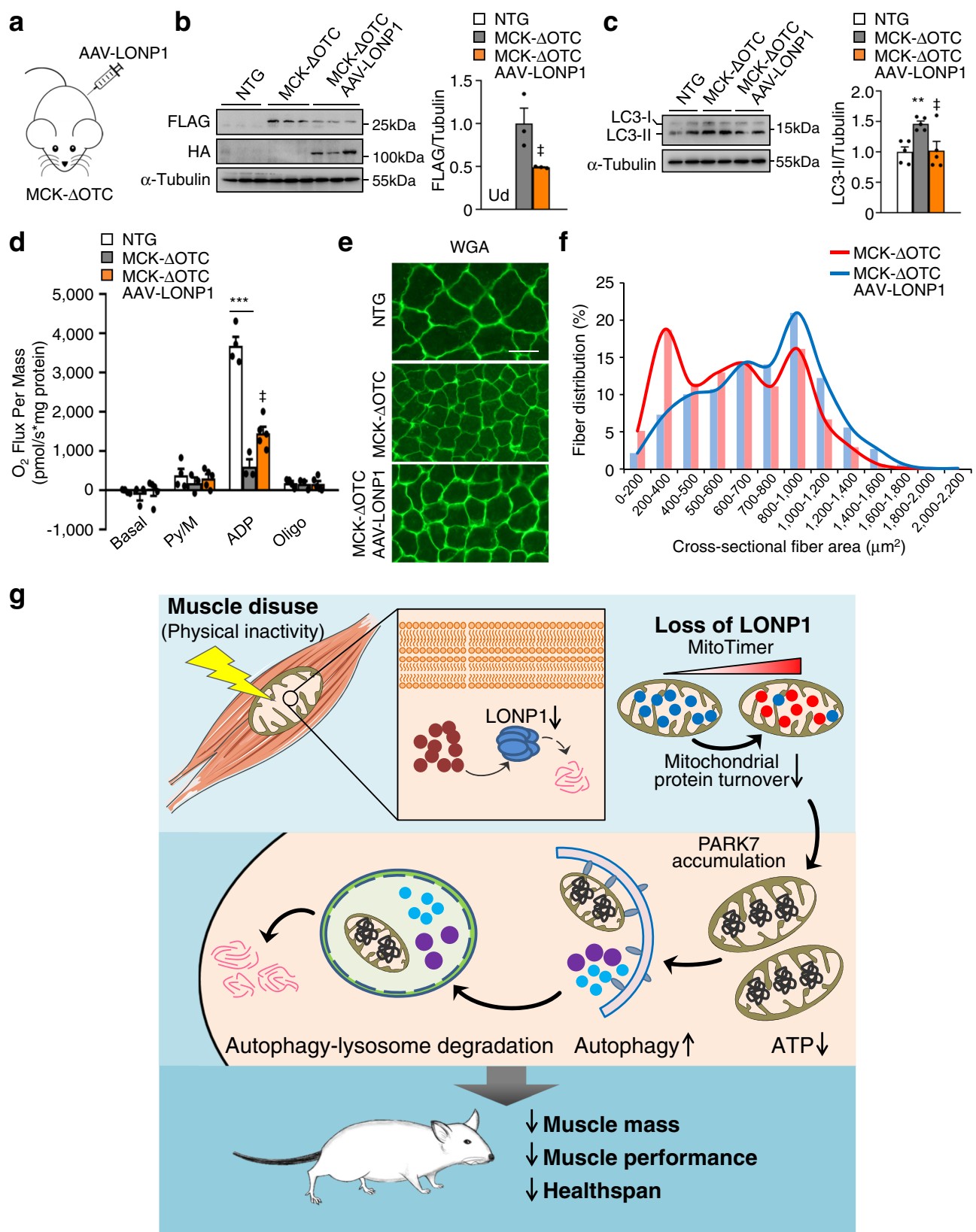

affecting PINK1 protein levels. This is different from a previous study on a siRNA-mediated LONP1 knockdown cell model[53]. Possible reasons that might account for the discrepancy could be the different techniques used for the LONP1 loss-of-function experiments or differences in primary myotubes and C2C12 cells. In addition, we did not observe a change in mitochondrial

mass and mtDNA content in LONP1 mKO muscle. It is possible that disturbed mitochondrial proteostasis triggers a unique mode of mitochondrial autophagy to selective degrade mitochondrial domains. This allows clearance of mitochondrial-retained damaged protein for lysosomal degradation, without affecting mitochondrial mass and mtDNA content. Indeed,

**Fig. 8 LONP1 preserves muscle function through maintaining mitochondrial proteostasis. a–f** MCK-ΔOTC mice were injected intraperitoneally on postnatal day 3 (P3) and day 5 (P5) with or without AAV expressing LONP1 (AAV-LONP1). Terminal studies were conducted 4 weeks after the injection of AAV. **b, c** Representative immunoblot analysis of muscle total protein extracts from indicated mice. **b** (Right) Quantification of the FLAG-ΔOTC/Tubulin signal ratios normalized (=1.0) to MCK-ΔOTC mice. Ud, undetectable. $n = 3$ mice per group. **c** (Left) Representative Western blot analysis of LC3-II protein in muscle total protein extracts from indicated mice. (Right) Quantification of the LC3-II/Tubulin signal ratios were normalized (= 1.0) to NTG controls. $n = 5$ mice per group. **d** Mitochondrial respiration rates were determined on mitochondria isolated from WV muscles of indicated mice using pyruvate as substrates. Pyruvate/malate (Py/M), ADP-dependent respiration, and oligomycin (oligo)-induced are shown. $n = 3$–5 mice per group. **e** Representative WGA (green) staining of GC muscle from indicated mice. Scale bar represents 50 μm. $n = 5$–6 mice per group. **f** Quantification of fiber size distribution in GC muscles from indicated mice. $n = 5$–6 mice per group. **g** Model of LONP1-mediated mitochondrial protein quality control in the regulation of skeletal muscle mass and strength. The schematic depicts a proposed model for disuse-associated loss of muscle LONP1 that controls mitochondrial function and skeletal muscle mass. Values represent mean ± SEM; for (**b**), ‡$P < 0.05$ versus MCK-ΔOTC determined by two-tailed unpaired Student's $t$ test; for (**c**) and (**d**), **$P < 0.01$, ***$P < 0.001$ versus corresponding controls, ‡$P < 0.05$ versus MCK-ΔOTC, determined by one-way ANOVA coupled to a Fisher's LSD post-hoc test; Source data are provided as a Source Data file.

previous studies have also reported that dysfunctional mitochondrial subdomains can be segregated from mitochondria and degraded by autophagy under stress[54].

In summary, we have revealed a role of LONP1-dependent mitochondrial protein quality control in safeguarding mitochondrial health and skeletal muscle mass and strength, unraveling an unexpected functional link between mitochondrial protein quality and muscle mass maintenance upon muscle disuse. The identification of LONP1 as a regulator of mitochondrial function and skeletal mass holds promise for translational applications in preventing muscle loss and weakness.

## Methods

**Animal studies**. All animal studies were conducted in strict accordance with the institutional guidelines for the humane treatment of animals and were approved by the IACUC committees at the Model Animal Research Center (MARC) of Nanjing University (Approval No. GZJ07). Male C57BL/6 J wild-type mice were from GemPharmatech Co., Ltd (Jiangsu, China). Generation of *Lonp1*f/f mice has been described elsewhere[55]. Mice were back-crossed to the C57BL/6 J background for more than 6 generations. To generate mice with a muscle-specific disruption of the *Lonp1* allele, *Lonp1*f/f mice were crossed with mice expressing Cre recombinase under the control of human skeletal actin (HSA) promoter (Jackson Laboratory; stock no. 006139) to achieve muscle-specific deletion of *Lonp1* (LONP1 mKO). To generate mice with muscle-specific overexpressing a mitochondrial-localized mutant OTC (MCK-ΔOTC), a cDNA encoding the rat mutant *Otc* gene was cloned into the EcoRV site downstream of the mouse *Mck* gene promoter (kind gift of E. N. Olson, University of Texas Southwestern). The transgene was linearized with XhoI and SacII digestion and microinjected into C57BL/6 J embryos by the transgenic mouse facility at the Model Animal Research Center of Nanjing University. Transgenic mice were identified by PCR amplification of a 552-bp product using primers specific for the component of the *Mck* gene promoter (5'-GGAACCAGTGAGCAAGTCAG) and *ΔOtc* (5'-GTGAGACCTTTGAGA-GAGCC). The CAG-CAT-MitoTimer reporter mice were kindly provided by Dr. Zhen Yan (University of Virginia). CAG-CAT-MitoTimer mice were crossed with HSA-Cre or LONP1 mKO mice to obtain HSA-Cre/MitoTimer or LONP1 mKO/MitoTimer mice to monitor the overall protein turnover status of skeletal muscle mitochondria. The mt-Keima Tg mice obtained from Jackson Laboratory (stock no. 028072) have been described previously[56]. To monitor the status of skeletal muscle mitochondrial autophagy, mt-Keima Tg mice were crossed with LONP1 mKO or MCK-ΔOTC mice to obtain LONP1 mKO/mt-Keima and MCK-ΔOTC/mt-Keima mice respectively. For chloroquine (CQ) treatment, CQ was administered intraperitoneally (i.p.) (50 mg/kg) to WT and LONP1 mKO mice every other day for 20 days. For rapamycin treatment, rapamycin was administered via i.p. (8 mg/kg/day) to mice for 4 weeks. Male offspring were genotyped and mice at the age of 2 weeks to 40 weeks were used. The animals were maintained with free access to pellet food and water in plastic cages at 21 ± 2 °C and kept on a 12 h light–dark cycle. All mice are harbored in the specific pathogen-free facility in Nanjing University. Littermate controls were used in all cases.

**Human studies**. Human supraspinatus muscle samples were collected from rotator cuff tear patients who underwent rotator cuff repair surgery. Clinical ethical approval in comply with the Declaration of Helsinki was obtained from the Ethical Committee of Subei People's Hospital of Jiangsu Province, Yangzhou, China (Clinical ethical approval: 2021ky-001). A total of 9 patients with rotator cuff tear were recruited for this study voluntarily. Details of the research project and biopsy collection were explained to all subjects before entering into the study. Informed consent was obtained from all subjects. A previous study has reported a reproducible and reliable method to define supraspinatus muscle atrophy with the help

of MRI[27]. Specifically, by calculating the occupation ratio which is the ratio between the surface of the cross-section of the muscle belly and that of the supraspinatus fossa, it allowed a reliable measurement of supraspinatus muscle atrophy, and the muscle with a ratio greater than 0.60 can be considered as normal or non-atrophied, while values below 0.60 suggest muscle atrophy[27]. We thus evaluated the atrophy of the supraspinatus muscle using the MRI-based method[27]. Briefly, MRI studies were carried out on all patients. MRI images were obtained with a ITesla unit (3.0 T Magnetom Trio, Siemens Healthcare, Erlangen, Germany). The oblique-sagittal images were taken parallel to the plane of the glenoid fossa, every 4 mm from the great tuberosity to the medial aspect of the scapular fossa. The quantitative analysis was performed on the spin-echo T1-weighted oblique-sagittal images (repetition time: 433 ms, echocardiographic time: 42 ms, field-of-view: 200 mm, matrix: 512 × 512, number of acquisitions: 3). The supraspinatus muscles were assessed using the most lateral oblique-sagittal section, which appears as "Y-shaped view" and we calculated the occupation ratio of the supraspinatus fossa by the muscle belly as previously described[27]. Details on subject characteristics are provided in Supplementary Table 1. Based on the occupation ratio, we divided the patients into two groups (non-atrophy group, mean ratio 0.76; atrophy group, mean ratio 0.50). After confirmation for full-thickness rotator cuff tear by arthroscopy, supraspinatus muscle biopsies were taken at 2 cm proximal zone of muscle–tendon junction during rotator cuff repair surgery, cleaned, and flash frozen in liquid nitrogen for protein isolation.

**Denervation and immobilization**. Denervation and immobilization were performed as previously described[24,51]. Eight-week-old male WT and LONP1 mKO mice (5–6 mice per group) were used. For denervation, in brief, a 5 mm segment of the sciatic nerve was surgically removed from the right hindlimb, and the non-denervated hindlimb served as controls. For immobilization, a splint was made from a cut 1.5 ml microfuge tube without a cap and a metal paperclip fixed by Velcro loop. Mouse right hindlimb was inserted into the splint, then the hindfoot and the end of paperclip were wrapped together with Velcro loop. This procedure prevents knee bending, and the non-immobilized hindlimb served as controls. Muscles were removed at 5, 10, or 15 days post denervation/immobilization and frozen in liquid nitrogen for subsequent analyses.

**Measuring in vivo mitochondrial autophagy**. For in vivo mitochondrial autophagy measurements in skeletal muscle, 2–6-week-old male mice (3 mice per group) were used. Single muscle fibers were isolated from fresh tissues. Fluorescence of mt-Keima was imaged in two channels via two sequential excitations (488 nm, green; 561 nm, red) using a Zeiss LSM880 confocal microscope (Carl Zeiss MicroImaging). Identical acquisition parameters were set for all samples in the same experimental condition. The mean intensity of mt-Keima signal was calculated using the Zeiss ZEN software.

**Measuring in vivo mitochondrial protein turnover**. In vivo MitoTimer measurements in skeletal muscle were conducted as previously described[57]. Two-to-six-week-old male mice (3 mice per group) were used. Fresh muscles were fixed straightly with 4% PFA for 2 h at 4 °C. Single muscle fibers were separated with microforceps under stereomicroscope. Fluorescence of MitoTimer was imaged in two channels via two sequential excitations (488 nm, green; 561 nm, red) using a Zeiss LSM880 confocal microscope (Carl Zeiss MicroImaging). Identical acquisition parameters were set for all samples in the same experimental condition. The mean intensity of MitoTimer signal was calculated using ImageJ (version 1.51, https://imagej.nih.gov/) software.

**Exercise stress test**. Mice were acclimated (run for 9 min at 10 m/min followed by 1 min at 20 m/min) to the treadmill for 2 consecutive days prior to the experimental protocol. Low intensity (endurance) exercise studies were conducted as described previously[46,58,59]. Fed mice were run for 10 min at 10 m/min followed by a constant speed of 20 m/min until exhaustion. Tail blood was taken after

exercise and measured for lactate (Lactate Scout, Senelab, Germany) according to the manufacturer's instruction.

Respiratory exchange ratios (RER) during exercise were determined as described previously[46,58]. Briefly, mice were placed in an enclosed treadmill attached to the Comprehensive Laboratory Animal Monitoring System (CLAMS) (Columbus Instruments) for 10 min at a 0° incline and 0 m/min. To determine maximal exercise capacity, the mice were subjected to a high-intensity exercise (wind sprints) test consisting of an increasing speed every 2 min at 0° inclination until exhaustion. The increasing speeds used in the protocol were 10, 14, 15, 16, 17, 18, 19, 20, 21, 22, 24, 26, 28, 30, 32, 34, 36, 38, 40, and 42 m/min. Measurements were collected before the exercise challenge and throughout the challenge.

**Muscle-grip strength measurements.** The grip strengths of forelimbs and all limbs were measured by Grip Strength Test (BIOSEB; EB Instruments) according to the instructions of the manufacturer. Eight-week-old male mice (8-12 mice per group) were used. Mice were allowed to hold on to a metal grid with four paws or only fore paws and were gently pulled backwards by the tail until the animals could no longer hold the grid. The average values of three tests were used to represent the muscle-grip strength for each mouse.

**Muscle tetanic contraction force measurements.** Eight-week-old male mice (3–4 mice per group) were used. EDL muscle were excised, and nonabsorbable silk surgical suture was tied to the tendons of muscles[60]. Muscles were incubated in oxygenated physiological salt solution (Krebs Buffer in mM: NaCl 118, KCl 4.7, CaCl₂ 2.5, MgCl₂ 1.2, NaH₂PO₄ 1, NaHCO₃ 25, glucose 11.1) at 37 °C. To obtain the contractile properties, the muscles were mounted between a force transducer and an adjustable hook (1200 A; Aurora Scientific) at a resting tension (L0). At the start of each experiment the muscle length and stimulated current were adjusted to yield the maximum force. Tetanic contraction was controlled by Dynamic Muscle Control software (Aurora Scientific) according to a standard experimental scheme to obtain the force output data from the stimulated muscle. Tetanic contraction was induced with the following specifications: initial delay, 0.1 s; frequency, 120 Hz; duration, 0.4 s.

**Histological analyses.** Mice muscle tissues were frozen in isopentane that had been cooled in liquid nitrogen. Tissue sections were stained with hematoxylin and eosin (Sigma-Aldrich) according to the standard protocol. Wheat germ agglutinin (WGA) staining was performed using FITC-conjugated WGA (Sigma-Aldrich, #L4859). Quantification of cross-sectional area of the myofibers was performed with NIH ImageJ software. Immunofluorescence (IF) stains were conducted as previously described[58,59]. The muscle fibers were stained with antibodies directed against MHC1 (BA-D5, catalog AB 2235587; Developmental Studies Hybridoma Bank) or MHC2b (BF-F3, catalog AB 2266724; Developmental Studies Hybridoma Bank). Immunohistochemistry was performed using the UltraSensitiveTM SP (Rabbit) IHC Kit (Maxim) according to the manufacturer's instructions using TFEB antibodies (A303-673A, Bethyl Laboratories). DAB Staining Kit (DAB-0031, Maxim) was used according to the manufacturer's instructions for the developing.

**X-ray.** Mice spine X-ray images were determined via Cabinet X-ray irradiator (Faxitron) according to the instructions of the manufacturer.

**In vivo protein synthesis rate measurements.** In vivo protein synthesis rates were measured via the SUnSET technique. Two-week-old male WT and LONP1 mKO mice (4–6 mice per group) were intraperitoneal injected with 0.04 mmol/g puromycin[61]. Thirty minute post-injection, muscles were collected and frozen in liquid Nitrogen for immunoblot analysis using anti-puromycin antibody (clone 12D10, Sigma).

**Isobaric tags for relative and absolute quantification (iTRAQ) mass spectrometry.** 100 µg mitochondrial proteins were reduced by adding Tris (2-carboxyethyl) phosphine (TCEP, AB Sciex, USA) to final concertation 5 mM for 1 h at 55 °C. Alkylation was performed using 6.25 mM S-Methyl methanethiosulfonate (MMTS, AB Sciex, USA) for 30 min at room temperature in darkness. Prior to trypsin digestion, the reduced and alkylated proteins were transferred to 10 K filter (Vivacon, USA) followed by centrifugation at 9560 g for 30 min to remove solvent and then wash three times with 0.5 M tetraethylammonium borohydride (TEAB, AB Sciex, USA). Trypsin (Promega, USA) was subsequently added into the filter at an enzyme: protein mass ratio of 1:50 for the first digestion overnight, and 1:100 enzyme: protein mass ratio for a 4 h digestion at 37 °C.

The resulting peptides were labeled with iTRAQ Reagent-4 plex Multiplex Kit (AB SCIEX, USA) according to the manufacturer's instructions. The two independent mitochondria samples from WT muscles were labeled with iTRAQ tag 113 and 114, and the two independent mitochondria samples from LONP1 mKO muscles were labeled with iTRAQ tag 115 and 116. The four samples were combined together and separated into 42 fractions by high-performance liquid chromatography (HPLC) system. The eluted 42 fractions were then combined into 16 samples for LC-MS/MS analysis. MS data acquisition was performed by LC-MS/MS analysis using a TripleTOF 5600+ System (AB SCIEX, Concord, Ontario, Canada) coupled with a NanoLC.2D (Eksigent Technologies). Samples were chromatographed using a 120 min gradient from 2 to 80% (mobile phase A 0.1% (v/v) formic acid, 2% (v/v) acetonitrile; mobile phase B 0.1% (v/v) formic acid, 98% (v/v) acetonitrile) after direct injection onto a nanoLC Column, 3C18-CL, 75 µm*15 cm (Eksigent Technologies). The gradient was comprised of an increase from 2% to 22% solvent B over 60 min, 22% to 35% B in 18 min and climbing to 80% B in 6 min then holding at 80% B for the last 6 min, all at a constant flow-rate of 300 nL/min. MS1 spectra were collected in the range 350–1500 $m/z$ for 250 ms. The 50 most intense precursors with charge state 2–5 were selected for fragmentation, and MS2 spectra were collected in the range 100–2000 $m/z$ for 100 ms; precursor ions were excluded from reselection for 15 s.

The original MS data were submitted to ProteinPilot Software (version 4.5, AB SCIEX) with Paragon Algorithm (4.5.0.0 1654) for data analysis. MS/MS data were searched against Mus musculus in the UniProt database (April 9, 2016, containing 160,566 sequences, http://www.uniprot.org/proteomes/UP000005640). The following search parameters were used: the instrument was TripleTOF 5600, iTRAQ quantification, cysteine modified with MMTS; biological modifications were selected as ID focus, quantitate, trypsin digestion, bias correction, and background correction were checked for protein quantification and normalization. False discovery rate (FDR) thresholds for both protein and peptide were specified at 1%. If multiple peptides were detected, the software will automatically select the peptides fitting in its selection criteria for protein quantification, most likely using their average to generate fold change information among different samples. To identify the differences between the LONP1 mKO group and the WT group, we specifically defined protein with fold change ≥ 1.2 as upregulated, and fold change ≤0.83 as downregulated. The waterfall plot was generated by using R software (Version 4.1.1) and ggplot2 package. For heatmap analysis, the filtered data sets were uploaded into Morpheus (https://software.broadinstitute.org/morpheus).

**Mitochondrial respiration studies.** Minced muscles were homogenized with a glass dounce on ice, and then centrifuged suspension twice at 600 g for 15 min at 4 °C. The resulting supernatants were carefully transferred to a new tube and centrifuged at 8,000 g for 15 min at 4 °C and the resulting supernatants were discarded. The pellet containing the mitochondria was washed twice and centrifuged at 8,000 g for 15 min at 4 °C before resuspension.

Mitochondrial respiration rates were measured in muscle mitochondria or primary myotubes with pyruvate or succinate as substrates as described previously[46,58,59]. In brief, muscle mitochondria were resuspended in buffer Z (105 mM potassium 2-[N-morpholino]-ethanesulfonic acid, 30 mM KCl, 10 mM KH₂PO4, 5 mM MgCl₂, 5 mg/ml BSA, 1 mM EGTA, pH7.4) at 37 °C and in the oxygen concentration range 220-150 nmol O₂/ml in the respiration chambers of an Oxygraph 2 K (Oroboros Inc., Innsbruck, Austria). Following measurement of basal, pyruvate (10 mM)/malate (5 mM) or succinate (5 mM)/rotenone (10 µM) respiration, maximal (ADP-stimulated) respiration was determined by exposing the mitochondria to 4 mM ADP. Uncoupled respiration was evaluated following the addition of oligomycin (1 µg/mL). Mitochondrial respiration measurements in myotubes were performed as follows. After basal oxygen consumption was recorded for 10 min, oligomycin (1 µg/mL) was injected. Then, FCCP (0.5 µM steps) was added to induce maximal respiration. Finally, rotenone (0.5 µM) was injected followed by antimycin-A (2.5 µM). Respiration rates were determined and normalized to mitochondrial protein contents or cell number using the Datlab 5 software (Oroboros Inc., Innsbruck, Austria), the data were expressed as "pmol O₂ s$^{-1}$ mg mito protein$^{-1}$" or "pmol O₂ s$^{-1}$ 10$^{-6}$ cells".

**Blue-native polyacrylamide gel electrophoresis.** Blue-native polyacrylamide gel electrophoresis (BN-PAGE) analyses were performed as described previously[62]. 250 µg mitochondria isolated as described above were resuspended in solubilization buffer (50 mM NaCl, 50 mM imidazole, 2 mM 6-aminohexanoic, and 1 mM EDTA, pH 7.0). Then, the mitochondria were incubated with 20% digitonin on ice for 10 min. After centrifugation at 20,000 × g for 30 min at 4 °C, the supernatants were collected. The samples were mixed with 50% glycerol and 5% Coomassie G250 and subjected to 3.5–13% blue-native PAGE gel for electrophoresis at 4 °C. After the native gel electrophoresis was conducted at 100 V for 30 min, cathode buffer B (50 mM Tricine, 7.5 mM imidazole, 0.02% Coomassie brilliant blue G-250) was changed to cathode buffer B/10 (50 mM Tricine, 7.5 mM imidazole, 0.002% Coomassie brilliant blue G-250) and the running continued at 15 mA for about 3 h. The gels were either stained for in-gel activities or electroblotted on PVDF membranes for immunodetection. For in-gel activity analysis, the gels were incubated in complex I (2 mM Tris·HCl, pH 7.4, 0.1 mg/ml NADH, 2.5 mg/ml Nitro Blue Tetrazolium (NBT)) or complex IV (50 mM phosphate buffer pH 7.4, 0.5 mg/mL diaminobenzidine (DAB), 1 mg/mL cytochrome c) substrate solution. The reactions were stopped with 10% acetic acid. Then the gels were washed with water and scanned.

**RNA-Seq studies.** Total RNA was isolated from the gastrocnemius muscle of 6-week-old male LONP1 mKO, and WT control mice using RNAiso Plus (Takara Bio). RNA-seq using Illumina HiSeq 4000 was performed by Beijing Novogene Bioinformatics Technology Co., Ltd. Two independent samples per group were analyzed. Paired-end, 150 nt reads were obtained from the same sequencing lane.

Transcriptome sequencing libraries averaged 42 million paired reads per sample, with 86.9% alignment to the mouse genome (UCSC mm10). The sequencing reads were then aligned to the UCSC mm10 genome assembly using TopHat 2.0.14 with the default parameters. Fragments Per Kb of exon per Million mapped reads (FPKM) were calculated using Cufflinks 2.2.1. The criteria for a regulated gene were a fold change greater than 1.5 (either direction) and a significant $P$-value (< 0.05) versus WT. The RNA-seq data have been deposited in the NCBI Gene Expression Omnibus and are accessible through GEO Series accession number GSE166071.

**RNA analyses**. Total RNA was extracted from the entire gastrocnemius or soleus muscle using RNAiso Plus (Takara Bio)[63]. The purified RNA samples were then reverse transcribed using the PrimeScript RT Reagent Kit with gDNA Eraser (Takara Bio). Real-time quantitative RT-PCR was performed using the ABI Prism Step-One system with Reagent Kit from Takara Bio. Specific oligonucleotide primers for target gene sequences are listed in Supplementary Table 2. Arbitrary units of target mRNA were corrected to the expression of $36b4$.

**Transmission electron microscopy**. Mice were euthanized and perfused with sodium phosphate buffer (PB, 100 mM, pH7.4) and pre-fixed solution (2.5% (vol/vol) glutaraldehyde, 1% paraformaldehyde in PB). Soleus muscle was dissected, cut into small pieces and fixed in the same pre-fixed solution overnight at 4 °C. After rinsing with PB, tissues were immersed in 0.2 M imidazole in PB for 15 min, and then post-fixed with 1% osmium tetraoxide in PB. After rinsing with high-purity water, the samples were stained with 1% aqueous lead at 4 °C overnight. Gradient dehydration was accomplished by incrementing the concentration of acetone and embedded in epoxy resin (60 °C for 24 h). Samples were sectioned using Leica EM UC7 and placed on copper grids. Images were taken on a FEI Tecnai G2 20 Twin electron microscope equipped with an Eagle 4k CCD digital camera (FEI; USA) in a double-blind manner.

**Autophagic flux analyses**. Autophagic flux was monitored in vivo muscle using colchicine (Sigma-Aldrich, C9754) as previously described[23]. WT and LONP1 mKO mice were i.p. injected with vehicle or with 0.4 mg/kg colchicine. The treatment was administered twice, at 24 h and at 12 h before muscle dissection. Myocytes were treated with 100 μM chloroquine (CQ) for 4 h to block autophagic flux[45].

**ATP assay**. The ATP levels were examined in muscle using a luciferase-luciferin ATP Assay Kit (Beyotime, China) following the manufacturer's instructions.

**Mitochondrial DNA analyses**. Mitochondrial DNA content was determined by SYBR Green analysis (Takara Bio). The levels of NADH dehydrogenase subunit 1 (mitochondrial DNA) were normalized to the levels of lipoprotein lipase (genomic DNA). The primer sequences are noted in Supplementary Table 2.

**Antibodies and immunoblotting studies**. Antibodies directed against LONP1 (15440-1-AP, 1:2,500 dilution), CLPP (15698-1-AP, 1:2,500 dilution), NDUFB8 (14794-1-AP, 1:1,000 dilution), ND1 (19703-1-AP, :1,000 dilution), SDHA (14865-1-AP, 1:1,000 dilution), UQCRC2 (14742-1-AP, 1:1,000 dilution), COX4 (11242-1-AP, 1:1,000 dilution), ATP5A (14676-1-AP, 1:1,000 dilution), MFN1 (13798-1-AP, 1:1,000 dilution), DRP1 (12957-1-AP, 1:1,000 dilution), PARK7 (11681-1-AP, 1:1,000 dilution), PINK1 (23274-1-AP, 1:1,000 dilution), GAPDH (60004-1-Ig, 1:1,000 dilution) were all from Proteintech; antibodies directed against OPA1 (612606, 1:1,000 dilution) and Tim23 (#611222, 1:1,000 dilution) were from BD Biosciences; antibodies directed against α-tubulin (bs1699, 1:5,000 dilution) were from Bioworld; antibodies directed against GFP (sc-9996, 1:1,000 dilution), MFN2 (sc-100560, 1:500 dilution) and Ubiquitin (sc-8017, 1:1,000 dilution) were from Santa Cruz; anti-LC3 antibody (NB100-2331, 1:1,000 dilution) and P62 (NBP1-48320, 1:1,000 dilution) were from Novus Biologicals; anti-Flag antibody (#F1804, 1: 1,000 dilution) and HA (H9658, 1:10,000 dilution) were from Sigma; anti-FOXO3A antibody (A0102, 1:1,000 dilution) was from Abconal, antibodies directed against p-FOXO3A Thr32 (#9464, 1:1,000 dilution), p-AKT Ser473 (#4060, 1:1,000 dilution), AKT (#9272, 1:1,000 dilution), p-mTOR Ser2448 (#5536, 1:1,000 dilution), p-mTOR Ser2481 (#2974, 1:1,000 dilution), mTOR (#2983, 1:1,000 dilution), p-S6K Thr389 (#9234, 1:1,000 dilution), S6K (#2708, 1:1,000 dilution), p-4EBP1 Thr37/46 (#2855, 1:1,000 dilution) and 4EBP1 (#9644, 1:1,000 dilution), LAMP1 (# 3243, 1:1,000 dilution), p-AMPKα Thr172 (#2535, 1:1,000 dilution) and AMPKα (#5831, 1:1,000 dilution) were from Cell Signaling Technology; anti-puromycin antibody (MABE343, clone 12D10, 1:5000 dilution) was from Sigma; anti-TFEB antibody (A303-673A, 1:200 dilution) was from Bethyl Laboratories; anti-MTCO1 antibody (ab14705, 1:1000 dilution) was from abcam; antibodies directed against MHC1 (BA-D5) and MHC2b (BF-F3) were purchased from the Developmental Studies Hybridoma Bank. Western blotting studies were performed as previously described[46,58,59]. Blots were normalized to α-tubulin. The total protein concentration was measure by BCA assay using Pierce BCA Assay Kit Protocol (ThermoFischer Scientific). Equal total protein was loaded to each lane.

**AAV injection**. AAVs for in vivo expression of GFP or Cre were generated and provided by the Rongsen Gene Technology Co., Ltd (Jiangsu, China). AAVs were diluted in 0.9% NaCl at $1 \times 10^{13}$ Vp/mL, and injected into muscles (30 μl per TA muscle). AAV-GFP was used as a control. To generate the recombinant AAV expressing the LONP1, PCR was performed using the primers: 5′-TCATTTTGGCAAAGAATTGGATCCGCCACCATGCGGCGAGCACAGGC-3′ and 5′-GAGGTTGATTATCGATAAGCTTTTAGGCATAATCTGGCA-CATCATAAGGGTATCTCTCCACGGCCAGTGCCTC-3′. The PCR products were then subcloned into an pAAV-CAG plasmid to produce the pAAV-CAG-LONP1 plasmid. AAVs were subsequently generated using packaging plasmids pAAV-helper and pAAV2/9 together with pAAV-CAG-LONP1 by Rongsen Gene Technology Co., Ltd (Jiangsu, China). Viruses were administered to mice (4–6 mice per group) by intraperitoneal injection at postnatal day 3 (P3) and day 5 (P5).

**Cell culture, RNAi experiments, and adenoviral infection**. GC muscles from both legs were removed. Minced tissue was digested in a collagenase/dispase/CaCl$_2$ solution for 1.5 h at 37 °C in a shaking bath. DMEM supplemented with 10% FBS (PPM) was added and samples were triturated gently before loading onto a Netwell filter (70 μm, BD). The cell suspension was pelleted at 950 $g$ for 5 min. Cells were then resuspended in PPM and plated on an uncoated plate for differential plating. Cell suspension (not-adherent) was centrifuged for 5 min at 950 $g$ and pellet was resuspended in Growth Medium (GM) (Ham's F-10 medium supplemented with 20% FBS and 2.5 ng/ml bFGF). Cells were plated on collagen-coated flasks for expansion. Cells were fed daily with GM. For differentiation, cells were washed with PBS and re-fed with 2% horse-serum/DMEM differentiation medium and re-fed daily. siRNAs (Genepharma) targeting mouse $Park7$ (5′CCAAAGGAGCAGAGGAGAGAUTT) were transfected into primary myoblasts at a final concentration of 50 nM using Lipofectamine 3000 Plus transfection reagent (Invitrogen) according to the manufacturer's instructions. Primary myoblasts were infected with an adenovirus overexpressing Cre or control virus as previously described[63], 12 h post-infection, cells were induced to differentiate into myotubes for 72 h prior to harvest.

**Myotube analysis**. Myotube diameter measurements were performed as described previously[64]. Average diameters of at least 100 myotubes were measured for each condition at three points along the length of the myotube. Three bright field images per dish were captured, randomized, and coded. The width was measured with the ImageJ software.

**Cycloheximide chase assay**. Primary myotubes were treated with cycloheximide (CHX; Selleck, S7418; 100 μg/mL) for 0 to 4 h. Cell lysates were then prepared by using a lysis buffer containing 2% SDS, followed SDS-PAGE and immunoblotting analyses with PARK7 antibody.

**Immunoprecipitation**. The $Park7$ and $Lonp1$ plasmid were generated by PCR amplification from cDNA of mouse $Park7$ and $Lonp1$, followed by cloning into the pcDNA3.1 and pCDH-CMV vectors. All constructs were confirmed by DNA sequencing. HEK293T cells were obtained from the American Type Culture Collection, and were cultured at 37 °C and 5% CO$_2$ in Dulbecco's modified Eagle's medium supplemented with 10% fetal calf serum, 1000 U/mL penicillin, and 100 mg/mL streptomycin. Transient transfections in HEK293T cells were performed using PEI Transfection Reagent (Polysciences) following the manufacturer's protocol. Whole lysate from HEK293T cells 48 h post-transfection was used for co-immunoprecipitation studies. HEK293 T cells were collected in lysis buffer (50 mM Tris, pH 7.5, 150 mM NaCl, 2 mM EDTA, 1.5% NP40, 1x Complete (Roche), and 1 mM PMSF). 1 μg of M2 anti-FLAG (Sigma) antibodies were incubated with extract and protein G-conjugated agarose beads. The immuno-precipitated proteins were analyzed by immunoblotting.

**Statistical analyses**. All mouse and cell studies were analyzed by two-tailed unpaired Student's $t$ test or one-way ANOVA coupled to a Fisher's least significant difference (LSD) post-hoc test when more than two groups were compared. Statistical analyses in human studies were performed using GraphPad Prism 8 software (Pearson's correlation test). No statistical methods were used to predetermine sample sizes, and sample size (range from $n = 3$ to $n = 10$) are explicitly stated in the figure legends. All data points were used in statistical analyses. Data represent the mean ± SEM, with a statistically significant difference defined as a value of $P < 0.05$.

**Reporting summary**. Further information on research design is available in the Nature Research Reporting Summary linked to this article.

## Data availability

The muscle LONP1 mKO RNA-seq data generated in this study have been deposited in the NCBI Gene Expression Omnibus database under accession number GSE166071. The proteomics data generated in this study have been deposited in PRIDE under accession code PXD029722. All other data supporting the findings of this study are available with the article, and can also be obtained from the authors. Source data are provided with this paper.

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

## Acknowledgements

The authors thank Dr. Zhen Yan (University of Virginia) for providing the MitoTimer reporter mice and Drs. Yong Liu (Wuhan University) and Zhongzhou Yang (Nanjing University) for reading the manuscript. This work was supported by grants from the National Natural Science Foundation of China (No. 91857105, 31771291, and 31922033 to Z.G. and 32071136 to T.F.), the Ministry of Science and Technology of China (National Key R&D Program of China 2018YFA0800700) and Natural Science Foundation of Jiangsu Province (BK20170014 and SWYY-002) to Z.G., Fundamental Research Funds for the Central Universities 090314380036 (to T.F.), 090314380031 and 090314380035 (to Z.G.).

## Author contributions

Z.X., T.F., and Q.G. contributed equally to this work and performed most of the experiments with assistance from D.Z., W.S., Z.Z., X.C., J.Z., L.L., L.X., Y.Y., Y.J., E.P. Order of co-first authors is based on the length of time spent on the project. Z.X., T.F., Q.G., and Z.G. designed experiments, discussed data, and wrote the manuscript. Y.C., X.P., L.F., M.Z., W.F., and B.L. contributed reagents and provided scientific insight and discussion. Z.G. supervised the work. All authors reviewed and contributed to the manuscript.

## Competing interests

The authors declare no competing interests.

## Additional information

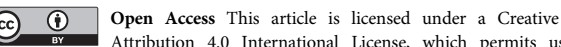

