## [Peer Review File · Nature Communications]

Reviewers' Comments:

Reviewer #1:

Remarks to the Author:

In this paper, Xu et al proved evidence of a role of Lonp1 in maintain quality control in skeletal muscle, and to prevent muscle mass and strength by characterizing the morphological and functional consequences of Lonp1 ablation in skeletal muscle.

The study is well designed and conducted in the first part, while I have some concerns regarding the experimental design and the conclusions drawn from the experiments regarding the functional consequences of Lonp1 KO on autophagy.

Major points

1) The aged LONP1 KO mice appear thinner than WT mice, as the weight loss was general and not only related to skeletal muscle. By the way, it is in apparent contrast with the claim that Lonp1 mKO mice consume less fat than WT. Is there an explanation for that?

2) LONP1 mKO mice are characterized by precocious aging, but I did not find any indications about lifespan. Is there an effect on that? This point should be indicated.

3) As the authors know very well, Lonp1 plays also a crucial role in maintaining mitochondrial DNA. As many mtDNA-related diseases show a deep impairment of skeletal muscles functions and homeostasis, have the authors checked mtDNA content in muscles from LONP1 KO mice? If not, mtDNA content has to be assessed in order to exclude that the effects they observed are due, at least in part, to this Lonp1 function.

4) Lonp1 interacts with, and degrades, proteins linked to regulation of mitophagic flux, such as PINK1. Although the role of PINK1/parkin in regulating mitophagy in skeletal muscle is debate, I think the authors have to provide evidence that this pathway (and DJ1/Park7) are not involved in changes they observed in autophagy. Notably, in the proteomic analysis shown, Park7 (recently shown to be degraded by Lonp1 at least in fibroblasts) is among the proteins strongly upregulated in KO mice (Figure 6d); could this suggest that Park7 is involved in increased autophagy (...mitophagy?) observed?

5) The experiments regarding DOTC overexpression are inconclusive, as far as Lonp1 role is concerned. These experiments show, in a convincing manner, that DOTC overexpression activates autophagy and causes muscle loss, but does not demonstrate that this effect can be modulated by Lonp1 (that is, when Lonp1 is absent, DOTC is overexpressed in skeletal muscle and activates autophagy). By the way, I did not find DOTC in the proteins present at higher levels in muscles from Lonp1 mKO mice. This casts some doubts on the idea that "defects in mitochondrial function can be due to the aberrant accumulation of mitochondrial retained proteins". My personal opinion – based upon similar observations present in other models – is that this could be the case, but it has to be demonstrated. Is there any effect on Lonp1 expression in these mice? What happens if Lonp1 is overexpressed in the same conditions? Is the effect on autophagy caused by DOTC dampened?

Minor points

-There are some typos and sentence not completely clear in the manuscript

Reviewer #2:

Remarks to the Author:

Mitochondrial function is known to play an integral role in maintaining the homeostasis of skeletal muscle in response to physiological and pathophysiological stresses. The proper functioning of the mitochondria is dependent on the activity of mitochondrial proteases that selectively degrade non-assembled, misfolded or damaged mitochondrial proteins. Lon protease homolog (LONP1) is an ATP-dependent mitochondrial protease that has been shown to mediate mitochondrial protein

quality.

In this manuscript, the authors assessed if the activity of LONP1 in the mitochondria is an important determinant of skeletal muscle function/integrity. Towards this end, they demonstrate that the expression of LONP1 decreases under conditions that cause muscle loss due to disuse. The importance of LONP1 in maintaining muscle mass/function was furthermore assessed by generating muscle-specific LONP1 knockout mice. They demonstrated, using these mice, that genetic ablation of LONP1 induces mitochondrial dysfunction, resulting in a loss of muscle mass and a decrease in muscle function. The authors suggest that these effects are due to the increased activation of the autophagy-lysosomal pathway. The results were also observed when LONP1 was knockdown in the skeletal muscle of *Lonp1f/f* mice using an adeno-associated virus (AAV) expressing the Cre-recombinase. The authors also observed similar phenotypes in mice overexpressing ornithine transcarbamylase (Δ OTC), a known target of LONP1. The authors conclude, from their studies, that LONP1-mediated mitochondrial protein quality control is an essential determinant of the integrity and function of skeletal muscle.

This is a very interesting manuscript that demonstrates, for the first time, the importance of LONP1 function in the mitochondria on skeletal muscle integrity/function. The authors, through the use of both in vivo and in vitro systems, have clearly demonstrated that LONP1 plays a prominent role in mediating muscle homeostasis. Despite the important contribution of these findings to the field, there are several issues that should be addressed. Indeed, in many instances, additional data will need to be included to further support the conclusions set forth by the authors. For example, although the authors have demonstrated that overexpression of Δ OTC in skeletal muscle induces mitochondrial dysfunction and loss of muscle mass, the authors should demonstrate either in vivo or, minimally, in primary skeletal myocytes, that this effect can be reversed by the overexpression of LONP1. This would help establish that the LONP1-mediated maintenance of skeletal muscle mass/function occurs, in part, due to degradation of Δ OTC. Other concerns/comments are outlined below:

Comments:

- 1) In Fig.1, the authors demonstrate that LONP1 protein levels decrease in muscle due to disuse caused by denervation or immobilization. The authors should assess both activation of the autophagy-lysosomal pathway and mitochondrial function in skeletal muscle under these conditions. Are both affected similarly to what was observed in muscle-specific LONP1 knockout mice?
- 2) The quantifications shown in Fig.1H demonstrating a decrease of LONP1 levels in human muscle from late diagnosis is not reflected in the western blot provided in Fig. 1G. Indeed, although the quantifications demonstrate a ~45% decrease in LONP1 levels, this effect is only apparent in 2 of the 6 samples in the western blot. The authors will need to provide a better western blot that more adequately reflects their quantifications.
- 3) The authors demonstrate in Fig. 2 that muscle-specific ablation of LONP1 results in a loss of muscle mass. It would be interesting to assess if this loss of muscle is exacerbated under conditions of disuse (either due to denervation or immobilization).
- 4) The authors should assess, in Fig. 2, if the effect of the genetic ablation of LONP1 on muscle metabolism is reflected by a shift in the composition of muscle fibers (oxidative versus glycolytic).
- 5) AMP-activated protein kinase (AMPK) is a well-known metabolic sensor that is activated in response to low ATP levels. The authors should assess, in Fig. 3, whether muscle-specific ablation of LONP1 induces activation of AMPK.
- 6) In Fig.4, the conclusion that LONP1 affects the autophagy- autophagy-lysosomal pathway should be supported by assessing additional autophagy-related markers including, among others, p62 and Beclin-1. This would help to further strengthen their conclusions.
- 7) The authors should demonstrate, in Fig. 5H, that infection of primary myoblasts isolated from *Lonp1f/f* mice with AD-Cre significantly reduces the expression of LONP1. They should also provide quantifications measuring the widths of the myotubes to conclude that depletion of LONP1 affects their size.
- 8) In Fig. 5I, the authors should assess if the impaired mitochondrial respiration observed in primary myotubes depleted of LONP1 is due to changes in the integrity of the mitochondrial respiratory chain complexes.

9) The knockdown of LONP1 has been previously shown to prevent muscle cell differentiation by regulating PINK1/Parkin-mediated mitochondrial remodeling (Huang et.al, Am. J. Physio. Cell Physio., 2020). Therefore, the authors should determine if the PINK/Parkin pathway is altered in the skeletal muscle of their LONP1 knockout mice and/or in their primary myotubes depleted of LONP1.

10) As stated above, additional markers of autophagy should be assessed in the experiments presented in Figs. 5J-L.

11) As stated above the authors should assess, in Fig.7, if the effect of overexpressing Δ OTC in skeletal muscle can be reversed by the overexpression of LONP1.

12) The authors conclude, from their histological analysis of muscles (shown in Supplementary Figure 2B), that knocking out LONP1 does not induce inflammation. No experiment data is however provided to support this. Indeed, these muscles appear to be marked by the increased infiltration of inflammatory cells. Therefore, the authors should prove experimentally that knocking out LONP1 did not induce an inflammatory state in these muscles.

Reviewer #3:

Remarks to the Author:

This paper documents the changes induced in skeletal muscle of Lonp1 ko mice under the control of the actin-dependent cre loxP endonuclease. Similar results were obtained in mice Lonp1(ff) treated with AAV-Cre in adulthood, indicating that the reported effects were present in mature muscle, as well as in early postnatal muscle. It is unclear at which stage the cre-actin endonuclease induces the Lonp1 ablation in skeletal muscle, the authors generally talk about "late stage" but a more detailed description of this issue is important to evaluate the precise timeframe of the genetic lesion induced by the cre-actin expression. The study on the LonP1 ko mice was prompted by the observation that LonP1 seems to be precociously and rather specifically decreased in skeletal muscle after denervation, or subjected to forced immobilization with deconditioning. These results are interesting, and shown in Figure 1 of the paper. Nevertheless, both denervation and disuse, as well as human hypotrophic changes induced in the muscle supraspinatus by chronic scapulo-humeral peri-arthritis, show a progressive reduction rather than complete abolition, of the protein, a condition which is very different from the LonP1 KO muscle-specific model subsequently created and investigated by the Authors. Therefore, a first question is whether the same or similar morphological and biochemical changes observed in LonP1 KO were investigated also in the first two models characterized by reduction of LonP1 amount, including the absence of increased ubiquitinated proteins, the modest if any induction of other mitochondrial proteases (CLPP1), and the lack of induction of the several gene products investigated as markers of atrophy or other functions in skeletal muscle. LonP1 is a crucial component of the proteostatic mechanism of mitochondria, it has been reported as an essential protease for the quality control of a number of mitochondrial systems, therefore it is not surprising that the ablation of this protein skeletal muscle can be deleterious. I am surprised by the absence of any evidence of activation of the ubiquitin-proteasome system when LonP1 is ablated, because this is a virtually universal mechanism to eliminate dysfunctional proteins, is partly linked to activation of autophagy, etc. Maybe this mechanism is not activated because most if not all changes in protein turnover induced by the ablation of LonP1 are affecting intramitochondrial proteins that are not processed by the ubiquitin-proteasome system? This is a possibility of course, but in the long-term the abolition of LonP1 should have some repercussion also on the general proteostasis of the cell, for instance by creating some effect on precursor proteins that should enter mitochondria through the outer and inner membrane-bound translocon proteins, etc. Therefore, evaluation of the absence of ubiquitination activation should be followed for a suitable amount of time. The mitochondrial energy metabolism seems to be clearly affected, by the observation of both in vivo defects induced by ablation of LonP1 on muscle strength and endurance, and the evidence of reduced oxygen consumption under ADP and substrate administration. This result points to a defect in the proteostasis of the mitochondrial respiratory chain complexes. Although no changes in the amount of subunits of the respiratory chain complexes is reported in the paper, the Authors should further investigate the integrity of the quaternary structures of the respiratory complexes by BN-GE-WB, with dodecyl-maltoside and, in order to evaluate the integrity of supercomplexes, digitonin as well. In addition, the activities of the single complexes should also be assessed, together with the measurement of ATP production and other bioenergetic features. The TEM results reported in the

paper clearly show profound alterations of the structure and integrity of the organelles, which can well justify the induction of autophagy, or mitophagy, aimed at eliminating the defective mitochondria. The experiments using KEIMA are interesting and convincing, as it is the measurement of the autophagic flux by colchicine/chloroquine sequence. Nevertheless, this set of elegant experiments should be completed by some independent approaches, including the traditional measurement of autophagic markers, including LC3-II etc. The concordant possible increase of lysosomal biogenesis should be proven by measuring Lamp1 and TFEB (phosphorylated and non-phosphorylated). The profound alteration of the organelles associated with LonP1 ablation can indeed trigger autophagy/mitophagy. This is entirely reasonable, but the crucial question is whether this is a specific mechanism associated to LonP1 ablation, or rather a general autophagic response against any disruptive condition affecting so dramatically the shape, structure, and presumably function of mitochondria? This question is not really addressed in the paper, despite the experiments done with a truncated variant of the ornithine carbamylase presented at the end of the work, the meaning of it is not completely clear. Importantly, as mentioned above, it would be interesting to test whether the organelle wasting observed with a complete abolition of LonP1 is also present in the denervated or immobilized muscle associated with reduced muscle mass and amount of LonP1. The point concerning the specificity of the autophagic process induced by LonP1 is an essential issue which is not really addressed by the paper, since LonP1 acts on a myriad of different substrates and an autophagic control of general mitochondrial homeostasis is reasonably triggered by a drastic insult such as the complete elimination of this important proteostatic element of mitochondria.

I am also wondering whether any effect towards a correction of the autophagic consequences of LonP1 ablation can be achieved by using antiautophagic agents such as brefeldin A, vincristine or chloroquine, and whether the morphological and biochemical alterations will persist under treatment with pro-autophagic agents such as rapamycin or rilmenidine.

Reviewer #4:

Remarks to the Author:

This manuscript by Xu et al. seeks to demonstrate that disuse leads to loss of muscle LONP1, which is responsible for mitochondrial dysfunction resulting in reduced skeletal muscle mass and strength. The authors use a number of cellular and animal models in a variety of experimental systems to provide proof for this. They also include one experiment done on human muscle biopsies to extend the findings to human. I feel that although the majority of the experiments related to loss of LONP1 are well done, the overall evidence does not really substantiate their claims. I have the following concerns with the manuscript:

Major issues:

1. The authors study samples obtained from the supraspinatus muscle from individuals diagnosed with rotator cuff injury. They claim that that the severity of this muscle loss depends upon the time from onset to the diagnosis. This is a really complicated system to use as there is not direct demonstration of muscle loss at all. In the early vs. late diagnosis groups, clinical scores that include a variety of measures including pain are provided and even those are highly overlapping across the groups. The location of the biopsy and the tissue recovered could skew the results greatly. Finally, the Western blots shown in Figure 1g are not convincing at all with differences observable only in two of the samples in the late diagnosis group - such small differences in light of the fact that Western blotting is not really a quantitative technique should be very carefully interpreted. I believe that the authors should obtain muscle biopsies from other, more extreme, clinical setting where the disuse and muscle loss is more clearly demonstrated.
2. The authors have done a proteomics experiments using iTRAQ-based labeling for quantitation. They describe that they considered proteins that displayed 1.2 fold-change. These experiments were done in duplicate in mice. I wonder why triplicate measurements were not done so that the authors could do robust statistical analyses. Also, the distribution of changes (such as in a waterfall plot) are not shown. Their cutoff of 20% change is also highly arbitrary and a really low cutoff to be able to make confident conclusions. Even across the replicate mice, their results shown in a highly diagrammatic fashion in Figure 6d are not really correlated. Since they describe

ornithine transcarbamylase (OTC) as a substrate of LONP1, I do not see OTC being mentioned as a molecule whose level was altered in this system. They do not discuss if they saw a single peptide or multiple peptides to derive their protein identification and quantitation. If multiple peptides were detected, what was the distribution of fold-changes across different peptides from the same protein? Finally, they should deposit their proteomics data to a public repository such as PRIDE.

3. I do not think overexpression experiments of a mitochondrial-localized mutant OTC are directly applicable to the role of LONP1. This is a highly artificial experiment, which only proves that overexpression of OTC in mitochondria in muscle causes muscle loss and not necessarily any connection to LONP1.

Minor issues:

1. The authors refer to an impact on mitochondrial "quality" - I think this phrase should be avoided and they should restrict their discussion to mitochondrial dysfunction or other more precise descriptions.

Itemized Responses to Reviewer #1

“In this paper, Xu et al proved evidence of a role of Lonp1 in maintain quality control in skeletal muscle, and to prevent muscle mass and strength by characterizing the morphological and functional consequences of Lonp1 ablation in skeletal muscle.

The study is well designed and conducted in the first part, while I have some concerns regarding the experimental design and the conclusions drawn from the experiments regarding the functional consequences of Lonp1 KO on autophagy.”

We sincerely appreciate the Reviewer’s critical and constructive review of our manuscript.

“The study is well designed and conducted in the first part”

Thank you for the positive comment.

Major points:

1. *“The aged LONP1 KO mice appear thinner than WT mice, as the weight loss was general and not only related to skeletal muscle. By the way, it is in apparent contrast with the claim that Lonp1 mKO mice consume less fat than WT. Is there an explanation for that?”*

We thank the reviewer. Our data indicate that LONP1 mKO mice consume less fat than WT mice during exercise based on the respiratory exchange ratio (RER) measurements, which is consistent with marked impaired muscle mitochondrial function in LONP1 mKO mice. As shown in original Fig. 2h and now in Fig. 2i, in LONP1 mKO mice, a rapid increase in RER values occurred at a very low speed during the exercise challenge, indicative of a switch to carbohydrates as the chief fuel. However, WT mice were able to exercise at much higher speeds before an increase in RER values was observed (Fig. 2i).

The reviewer has raised an interesting question. The aged LONP1 mKO mice are smaller, and we are interested in this, as we have demonstrated previously that mice lacking muscle mitophagy receptor FUNDC1 were protected against high-fat-diet-induced obesity despite reduced muscle fat utilization during exercise. Based on our previous studies and the published literature, the seemingly contradictory phenotypes may relate to the non-cell autonomous role of the muscle mitochondrial stress response (Fu et al., Cell Rep. 2018;23(5):1357-1372; Pereira et al., EMBO J. 2017;36(14):2126-2145). For instance, FGF21, a mitokine known to have favorable effect on systemic energy metabolism, is induced during muscle mitochondrial stress. We wish to further delineate the LONP1 signaling in regulating muscle mitochondrial stress response and systemic metabolism. This information has been added to the revised Discussion (page 26, line 576-586).

2. *“LONP1 mKO mice are characterized by precocious aging, but I did not find any indications about lifespan. Is there an effect on that? This point should be indicated.”*

We appreciate the Reviewer’s concerns. We followed a cohort of LONP1 mKO mice that show similar survival as respective WT littermates at least up to 1.5 years of age. This information was in the original Text (page 10, line 204-206).

-
3. *“As the authors know very well, Lonp1 plays also a crucial role in maintaining mitochondrial DNA. As many mtDNA-related diseases show a deep impairment of skeletal muscles functions and homeostasis, have the authors checked mtDNA content in muscles from LONP1 KO mice? If not, mtDNA content has to be assessed in order to exclude that the effects they observed are due, at least in part, to this Lonp1 function.”*

We thank the Reviewer. As shown in original Fig 3c and addressed in the original Discussion (page 27, line 612-619), we have performed quantitative RT-qPCR to determine mitochondrial DNA levels (original method, page 40-41, line 924-928). We did not observe a change in mtDNA content in the GC muscles of LONP1 mKO mice compared to WT controls (original Fig. 3c). These results suggest that loss of LONP1 alters muscle function without affecting mtDNA content.

4. *“Lonp1 interacts with, and degrades, proteins linked to regulation of mitophagic flux, such as PINK1. Although the role of PINK1/parkin in regulating mitophagy in skeletal muscle is debate, I think the authors have to provide evidence that this pathway (and DJ1/Park7) are not involved in changes they observed in autophagy. Notably, in the proteomic analysis shown, Park7 (recently shown to be degraded by Lonp1 at least in fibroblasts) is among the proteins strongly upregulated in KO mice (Figure 6d); ì could this suggest that Park7 is involved in increased autophagy (...mitophagy?) observed?”*

We appreciate the Reviewer’s insightful points. We agree with the Reviewer in that it would be interesting to test whether LONP1 directly interacts with and degrades proteins involved in autophagy. We have conducted additional experiments to address this intriguing question. As a first step, we demonstrated that PINK1 protein levels remained unchanged in LONP1 KO myotubes compared to WT controls (revised Fig. 6e), indicating that loss of LONP1 activates muscle cell autophagy without affecting PINK1 protein levels. Given that several published studies have suggested that PARK7 may regulate autophagy (Lee et al., *Autophagy*. 2018;14(11):1870-1885.; Xue et al., *Arch Biochem Biophys*. 2017;633:124-132; Krebiehl et al., *PLoS One*. 2010;5(2):e9367; Gao et al., *J Mol Biol*. 2012;423(2):232-48), we also determined whether PARK7 protein, whose expression was up-regulated in LONP1 mKO muscle mitochondria, is involved in mediating LONP1 regulation of autophagy. We have found that PARK7 is a LONP1 substrate in muscle cells, and PARK7 protein was stabilized in the absence of LONP1 in myotubes (revised Fig. 6g). Moreover, loss-of-function experiments in primary myotubes showed that siRNA knockdown of *Park7* reduced autophagy flux in LONP1 KO myotubes (revised Fig. 6i, j). Together, our new studies identified PARK7 as a LONP1 substrate in muscle cells, which, when knocked down, reduced autophagy flux in LONP1-deficient muscle cells.

The above information has been added as revised Fig. 6e-j (page 19, line 422-442) and addressed in the revised Discussion (page 27, line 602-612).

5. *“The experiments regarding DOTC overexpression are inconclusive, as far as Lonp1 role is concerned. These experiments show, in a convincing manner, that DOTC overexpression activates autophagy and causes muscle loss, but does not demonstrate that this effect can be modulated by Lonp1 (that is, when Lonp1 is absent, DOTC is overexpressed in skeletal muscle and activates*

autophagy). By the way, I did not find DOTC in the proteins present at higher levels in muscles from Lonp1 mKO mice. This casts some doubts on the idea that “defects in mitochondrial function can be due to the aberrant accumulation of mitochondrial retained proteins”. My personal opinion – based upon similar observations present in other models – is that this could be the case, but it has to be demonstrated. Is there any effect on Lonp1 expression in these mice? What happens if Lonp1 is overexpressed in the same conditions? Is the effect on autophagy caused by DOTC dampened?”

The Reviewer has raised an interesting and important point regarding Δ OTC overexpression in skeletal muscle. We agree with the Reviewer and have addressed this point experimentally.

We also think it is worth noting that the deletion mutant of ornithine transcarbamylase (Δ OTC) is a known protein degraded by LONP1 and a well-established model for studying mitochondrial proteostasis imbalance (Zhao et al., EMBO J. 2002;21(17):4411-9; Jin SM, Youle RJ. Autophagy. 2013;9(11):1750-7; Bezawork-Geleta et al., Sci Rep. 2015;5:17397; Quirós et al., Nat Rev Mol Cell Biol. 2015;16(6):345-59.). In fact, the physiological relevance of imbalanced mitochondrial proteostasis remains largely unclear. We showed that muscle-specific overexpressing mitochondrial-retained Δ OTC protein is sufficient to induce mitochondrial dysfunction, autophagy activation, and cause muscle loss and weakness, thus, we suggest that the accumulation of misfolded proteins in mitochondria is a physiological trigger of mitochondrial dysfunction and muscle loss is an important and novel aspect in this regard.

We have conducted additional experiments as suggested. Notably, it has been established that increased workload of unfolded proteins at the mitochondrial can trigger the activation of the mitochondrial proteases. Consistently, we have found that overexpression of Δ OTC triggered a mild induction of LONP1 protein in skeletal muscles (new Supplementary Fig. 9b), indicating a mitochondrial proteostasis stress in MCK- Δ OTC muscles. To determine whether overexpression of LONP1 could impact the muscle phenotypes in MCK- Δ OTC mice, we generated recombinant AAV2/9 to express LONP1 in MCK- Δ OTC muscles *in vivo* through systemic delivery to neonatal mice. Consistent with LONP1 degrading Δ OTC (Jin SM, Youle RJ. Autophagy. 2013;9(11):1750-7; Bezawork-Geleta et al., Sci Rep. 2015;5:17397.), overexpression of LONP1 resulted in a significant reduction of Δ OTC protein in MCK- Δ OTC muscles (new Fig. 8b). As predicted, we have found that overexpression of LONP1 not only reduced the amount of LC3-II, but also significantly increased the mitochondrial respiration capacity in MCK- Δ OTC muscles (new Fig. 8c, d). Moreover, quantification of muscle fiber size distribution revealed that overexpression of LONP1 also led to a shift toward larger myofibers in MCK- Δ OTC mice (new Fig. 8e, f). These results further suggest that LONP1 plays a crucial role in preserving muscle function through maintaining mitochondrial proteostasis.

The above information has been added as new Fig. 8a-f and new Supplementary Figure 9b (page 20, line 452-454, page 21-22, line 485-496) and addressed in the revised Discussion (page 24, line 538-545).

Minor point:

1. *“There are some typos and sentence not completely clear in the manuscript.”*

We thank the reviewer. We have tried our best to correct the typographical errors throughout the manuscript.

Itemized Responses to Reviewer #2

“Mitochondrial function is known to play an integral role in maintaining the homeostasis of skeletal muscle in response to physiological and pathophysiological stresses. The proper functioning of the mitochondria is dependent on the activity of mitochondrial proteases that selectively degrade non-assembled, misfolded or damaged mitochondrial proteins. Lon protease homolog (LONP1) is an ATP-dependent mitochondrial protease that has been shown to mediate mitochondrial protein quality.

*In this manuscript, the authors assessed if the activity of LONP1 in the mitochondria is an important determinant of skeletal muscle function/integrity. Towards this end, they demonstrate that the expression of LONP1 decreases under conditions that cause muscle loss due to disuse. The importance of LONP1 in maintaining muscle mass/function was furthermore assessed by generating muscle-specific LONP1 knockout mice. They demonstrated, using these mice, that genetic ablation of LONP1 induces mitochondrial dysfunction, resulting in a loss of muscle mass and a decrease in muscle function. The authors suggest that these effects are due to the increased activation of the autophagy-lysosomal pathway. The results were also observed when LONP1 was knockdown in the skeletal muscle of *Lonp1^{ff}* mice using an adeno-associated virus (AAV) expressing the Cre-recombinase. The authors also observed similar phenotypes in mice overexpressing ornithine transcarbamylase (Δ OTC), a known target of LONP1. The authors conclude, from their studies, that LONP1-mediated mitochondrial protein quality control is an essential determinant of the integrity and function of skeletal muscle.*

This is a very interesting manuscript that demonstrates, for the first time, the importance of LONP1 function in the mitochondria on skeletal muscle integrity/function. The authors, through the use of both in vivo and in vitro systems, have clearly demonstrated that LONP1 plays a prominent role in mediating muscle homeostasis. Despite the important contribution of these findings to the field, there are several issues that should be addressed. Indeed, in many instances, additional data will need to be included to further support the conclusions set forth by the authors. For example, although the authors have demonstrated that overexpression of Δ OTC in skeletal muscle induces mitochondrial dysfunction and loss of muscle mass, the authors should demonstrate either in vivo or, minimally, in primary skeletal myocytes, that this effect can be reversed by the overexpression of LONP1. This would help establish that the LONP1-mediated maintenance of skeletal muscle mass/function occurs, in part, due to degradation of Δ OTC. Other concerns/comments are outlined below:”

We wish to thank the Reviewer for the critical and constructive review. Our itemized responses are as follows:

“This is a very interesting manuscript that demonstrates, for the first time, the importance of LONP1 function in the mitochondria on skeletal muscle integrity/function. The authors, through the use of both in vivo and in vitro systems, have clearly demonstrated that LONP1 plays a prominent role in mediating muscle homeostasis.”

Thank you for the positive comment.

Comments:

1. *“In Fig.1, the authors demonstrate that LONP1 protein levels decrease in muscle due to disuse caused by denervation or immobilization. The authors should assess both activation of the*

autophagy-lysosomal pathway and mitochondrial function in skeletal muscle under these conditions. Are both affected similarly to what was observed in muscle-specific LONP1 knockout mice?"

We appreciate the Reviewer's insightful points. As suggested, we have conducted additional autophagy and mitochondrial respiration measurements in muscles following denervation. We have found that, as predicted, the red mt-Keima fluorescence signal was significantly induced, an indicator of mitochondrial autophagy, in EDL muscles of mt-Keima Tg mice at 5 days following denervation (new Supplementary Fig. 7a), which was accompanied by increased conversion of LC3-I to LC3-II and reduction of muscle mitochondrial respiration capacity (new Supplementary Fig. 7b, c). Thus, our data suggest that denervation-induced muscle disuse leads to suppressed expression of LONP1 with increased muscle autophagy and mitochondrial dysfunction, which is consistent with our observations in LONP1 mKO muscles. These new data have been added to new Supplementary Figure 7 (page 17, line 381-392).

2. *"The quantifications shown in Fig. 1H demonstrating a decrease of LONP1 levels in human muscle from late diagnosis is not reflected in the western blot provided in Fig. 1G. Indeed, although the quantifications demonstrate a ~45% decrease in LONP1 levels, this effect is only apparent in 2 of the 6 samples in the western blot. The authors will need to provide a better western blot that more adequately reflects their quantifications."*

We thank the Reviewer. We have investigated further, the variations in LONP1 protein levels could reflect that some of our original human muscle samples from late diagnosis were non-atrophied, and we have conducted new experiments to examine the regulation of LONP1 protein in supraspinatus muscle samples from new patients with and without muscle atrophy. Please also see our detailed response to point #1 raised by Reviewer #4. In brief, We have evaluated the atrophy of the supraspinatus muscle using the MRI-based method (Thomazeau et al., Acta Orthop Scand. 1996;67(3):264-8.), and we calculated the occupation ratio of the supraspinatus fossa by the muscle belly. Based on the occupation ratio and previous report (Thomazeau et al., Acta Orthop Scand. 1996;67(3):264-8.), we divided the patients into two groups (non-atrophy group, mean ratio 0.76; atrophy group, mean ratio 0.50) (revised Supplementary Table 1). We found that the protein levels of LONP1 and CLPP were significantly reduced in human muscles from the atrophy group compared to non-atrophy group (revised Fig. 1g, h), In addition, both LONP1 and CLPP proteins exhibited a significant positive correlation with occupation ratios (new Fig. 1i). In contrast, the levels of OPA1 protein were not different between atrophy and non-atrophy groups, and neither exhibited a significant correlation with occupation ratio (revised Fig. 1g and revised Supplementary Figure 1f, g). Our new data suggest that muscle loss in humans is associated with decreased mitochondrial proteases. The new data have been added to the revised Fig. 1g-i and revised Supplementary Figure 1f, g (page 8-9, line 159-173, Methods, page 30-31, line 662-689).

3. *"The authors demonstrate in Fig. 2 that muscle-specific ablation of LONP1 results in a loss of muscle mass. It would be interesting to assess if this loss of muscle is exacerbated under conditions of disuse (either due to denervation or immobilization)."*

We thank the Reviewer. Following the Reviewer's suggestions, we have conducted additional denervation studies in WT and LONP1 mKO mice. We have found that loss of LONP1 exacerbated denervation-induced muscle atrophy. LONP1 mKO mice showed more pronounced myofiber atrophy and muscle loss in GC and TA muscles in response to denervation (revised Fig. 2b-e). These new results have been added to the revised Fig. 2b-e (page 10, line 206-210).

4. *"The authors should assess, in Fig. 2, if the effect of the genetic ablation of LONP1 on muscle metabolism is reflected by a shift in the composition of muscle fibers (oxidative versus glycolytic)."*

We thank the Reviewer. Interestingly, we found that the reduced exercise endurance in LONP1 mKO mice were not due to reduce in slow-twitch type I muscle fibers. Indeed, surprisingly, MHC1 immunofluorescence staining revealed a striking increase in the number of type I fibers in LONP1 mKO GC muscles compared to WT controls (revised Supplementary Fig. 2i). Moreover, expression of the gene encoding the major slow-twitch type I myosin isoform MHC1 (*Myh7* gene) and slow twitch troponin genes (*Tnni1*, *Tnnc1*, and *Tnnt1*) was increased in LONP1 mKO GC muscles (revised Supplementary Fig. 2j). In contrast, expression of the fast-twitch type II myosin isoform MHC2b (*Myh4* gene) and fast-twitch troponin genes (*Tnni2*, *Tnnc2*, and *Tnnt3*) was reduced in LONP1 mKO GC muscles (revised Supplementary Fig. 2j). These results suggest a dissociation between mitochondrial function and type I fiber program in LONP1 mKO mice muscles. The observed increase in type I fibers in LONP1 mKO mice are consistent with previous report that mitochondrial energetic deficiency can trigger compensatory muscle fiber type switching (Zechner et al., Cell Metab. 2010;12(6):633-42.). These new data have been added to the revised Supplementary Figure 2i, j (page 11, line 227-236) and addressed in the revised Discussion (page 25, line 551-555).

5. *"AMP-activated protein kinase (AMPK) is a well-known metabolic sensor that is activated in response to low ATP levels. The authors should assess, in Fig. 3, whether muscle-specific ablation of LONP1 induces activation of AMPK."*

Thank you for this query. Intriguingly, the p-AMPK/AMPK ratio was significantly increased in LONP1 mKO muscles (revised Supplementary Figure 3e). The activation of AMPK is consistent with cellular sensing of relative energetic deficiency within LONP1 mKO myofibers. The new AMPK data have been added to the revised Supplementary Figure 3e (page 13, line 283-284).

6. *"In Fig.4, the conclusion that LONP1 affects the autophagy- autophagy-lysosomal pathway should be supported by assessing additional autophagy-related markers including, among others, p62 and Beclin-1. This would help to further strengthen their conclusions."*

We have added autophagy marker P62 in our autophagy flux analysis (revised Fig. 4i) as suggested. LONP1 mKO muscles showed higher colchicine-induced accumulation of P62 protein than WT controls (revised Fig. 4i). This supports an enhanced autophagy flux in LONP1 mKO muscles.

7. *"The authors should demonstrate, in Fig. 5H, that infection of primary myoblasts isolated from *Lonplff* mice with AD-Cre significantly reduces the expression of LONP1. They should also provide*

quantifications measuring the widths of the myotubes to conclude that depletion of LONP1 affects their size.”

As shown in the original Fig. 5k, l, and now in revised Fig. 5l, m, adenoviral Cre-mediated KO of *Lonp1* in myocytes resulted in marked reduced expression of LONP1 protein. As suggested, we have quantified the changes in myotube diameters (Methods, page 43-44, line 997-1001). Adenoviral Cre-mediated KO of *Lonp1* resulted in a decrease in myotube diameter (revised Fig. 5h). These results were added to the revised Fig. 5h (page 16, line 365-366).

8. *“In Fig. 5I, the authors should assess if the impaired mitochondrial respiration observed in primary myotubes depleted of LONP1 is due to changes in the integrity of the mitochondrial respiratory chain complexes.”*

We followed the Reviewer’s suggestion. We have analyzed the level of assembled mitochondrial respiratory chain complexes by BN-PAGE and subsequent Western blot analysis. We also found a decrease in the amount of fully assembled complexes IV in LONP1 KO myotubes relative to WT controls (revised Supplementary Figure 6e). These new data have been added to the revised Supplementary Figure 6e (page 17, line 370-372).

9. *“The knockdown of LONP1 has been previously shown to prevent muscle cell differentiation by regulating PINK1/Parkin-mediated mitochondrial remodeling (Huang et.al, Am. J. Physio. Cell Physio., 2020). Therefore, the authors should determine if the PINK/Parkin pathway is altered in the skeletal muscle of their LONP1 knockout mice and/or in their primary myotubes depleted of LONP1.”*

Notably, adenoviral Cre-mediated KO of *Lonp1* did not influence primary myotube differentiation (original Fig. 5h). In addition, we have added new data demonstrated that PINK1 protein levels remained unchanged in LONP1 KO myotubes compared to WT controls (revised Fig. 6e). These data suggest that LONP1 effects on muscle cell function without affecting PINK1 protein levels. This is in contrast to a previous study on a siRNA-mediated LONP1 knock down C2C12 model (Huang et.al, Am J Physiol Cell Physiol. 2020;319(6):C1020-C1028.). Possible reasons that might account for the discrepancy could be the different techniques used for the LONP1 loss-of-function experiments or differences in primary myotubes and C2C12 cells. The new data have been added to the revised Fig. 6e (page 19, line 430-431) and we have also added a discussion point to the revised Discussion (page 27, line 607-612).

10. *“As stated above, additional markers of autophagy should be assessed in the experiments presented in Figs. 5J-L.”*

We have added autophagy marker P62 in our autophagy flux analysis in primary myotubes (revised Fig. 5m) as suggested. We also found more increase of P62 protein upon CQ treatment in LONP1 KO myotubes compared to controls (revised Fig. 5m), confirming an enhanced autophagy flux in LONP1 KO myotubes.

11. *“As stated above the authors should assess, in Fig.7, if the effect of overexpressing Δ OTC in skeletal muscle can be reversed by the overexpression of LONP1.”*

The Reviewer has raised an interesting and important point regarding Δ OTC overexpression in skeletal muscle. We agree with the Reviewer and have addressed this point experimentally. We also think it is worth noting that the deletion mutant of ornithine transcarbamylase (Δ OTC) is a known protein degraded by LONP1 and a well-established model for studying mitochondrial proteostasis imbalance (Zhao et al., EMBO J. 2002;21(17):4411-9; Jin SM, Youle RJ. Autophagy. 2013;9(11):1750-7; Bezawork-Geleta et al., Sci Rep. 2015;5:17397; Quirós et al., Nat Rev Mol Cell Biol. 2015;16(6):345-59.). In fact, the physiological relevance of imbalanced mitochondrial proteostasis remains largely unclear. We showed that muscle-specific overexpressing mitochondrial-retained Δ OTC protein is sufficient to induce mitochondrial dysfunction, autophagy activation, and cause muscle loss and weakness, thus, we suggest that the accumulation of misfolded proteins in mitochondria is a physiological trigger of mitochondrial dysfunction and muscle loss is an important and novel aspect in this regard.

Following the Reviewer’s suggestion, we have conducted additional experiments to examine whether overexpression of LONP1 could impact the muscle phenotypes in MCK- Δ OTC mice. We generated recombinant AAV2/9 to express LONP1 in MCK- Δ OTC muscle *in vivo* through systemic delivery to neonatal mice. Consistent with LONP1 degrading Δ OTC (Jin SM, Youle RJ. Autophagy. 2013;9(11):1750-7; Bezawork-Geleta et al., Sci Rep. 2015;5:17397.), overexpression of LONP1 resulted in a significant reduction of Δ OTC protein in MCK- Δ OTC muscles (new Fig. 8b). As predicted, we have found that overexpression of LONP1 not only reduced the amount of LC3-II, but also significantly increased the mitochondrial respiration capacity in MCK- Δ OTC muscles (new Fig. 8c, d). Moreover, quantification of muscle fiber size distribution revealed that overexpression of LONP1 also led to a shift toward larger myofibers in MCK- Δ OTC mice (new Fig. 8e, f). These results further suggest that LONP1 plays a crucial role in preserving muscle function through maintaining mitochondrial proteostasis.

The above information has been added as new Fig. 8a-f (page 21-22, line 485-496) and addressed in the revised Discussion (page 24, line 538-545).

12. *“The authors conclude, from their histological analysis of muscles (shown in Supplementary Figure 2B), that knocking out LONP1 does not induce inflammation. No experiment data is however provided to support this. Indeed, these muscles appear to be marked by the increased infiltration of inflammatory cells. Therefore, the authors should prove experimentally that knocking out LONP1 did not induce an inflammatory state in these muscles.”*

We thank the Reviewer. We have conducted additional RT-qPCR analysis in LONP1 mKO muscles. We did not detect induction of major inflammation-associated genes (such as *F4/80*, *Cd68*, *Cd11c*, *Il6*, and *Tnfa*) in LONP1 mKO muscles. These new data are provided in a revised Supplementary Figure 2e (page 10, line 195-196).

Itemized Responses to Reviewer #3

We wish to thank the Reviewer for the critical and constructive review. Our itemized responses are as follows:

1. *“This paper documents the changes induced in skeletal muscle of Lonp1 ko mice under the control of the actin-dependent cre loxP endonuclease. Similar results were obtained in mice Lonp1(ff) treated with AAV-Cre in adulthood, indicating that the reported effects were present in mature muscle, as well as in early postnatal muscle. It is unclear at which stage the cre-actin endonuclease induces the Lonp1 ablation in skeletal muscle, the authors generally talk about "late stage" but a more detailed description of this issue is important to evaluate the precise timeframe of the genetic lesion induced by the cre-actin expression.”*

We appreciate the Reviewer’s concerns regarding the developmental onset of HSA-Cre-mediated LONP1 excision. We agree with the Reviewer and have addressed this point experimentally. Efficient postnatal deletion of LONP1 in skeletal muscle by HSA-Cre was verified by quantitative PCR analysis of LONP1 in mice at the age of 1, 7 and 14 days. Our data suggest that efficient ablation of LONP1 mediated by HSA-Cre did not occur until ~7 days after birth (revised Supplementary Fig. 2a), which is consistent with previous report (Cifuentes-Diaz et al., J Cell Biol. 2001;152(5):1107-14). This information has been added to revised Supplementary Fig. 2a (page 9, line 184-186).

2. *“The study on the LonP1 ko mice was prompted by the observation that LonP1 seems to be precociously and rather specifically decreased in skeletal muscle after denervation, or subjected to forced immobilization with deconditioning. These results are interesting, and shown in Figure 1 of the paper. Nevertheless, both denervation and disuse, as well as human hypotrophic changes induced in the muscle sopraspinatus by chronic scapulo-humeral periarthritis, show a progressive reduction rather than complete abolition, of the protein, a condition which is very differente from the LonP1 KO muscle-speciifc model subsequently created and investigated by the Authors. Therefore, a first question is whether the same or similar morphological and biochemical changes observed in LonP1 KO were investigated also in the first two models characterized by reduction of LonP1 amount, including the absence of increased ubiquitinated proteins, the modest if any induction of other mitochondrial proteases (CLPP1), and the lack of induction of the several gene products investigated as markers of atrophy or other functions in skeletal muscle.”*

We appreciate the Reviewer’s insightful points. As suggested, we have conducted additional experiments in denervation-induced muscle loss mice model. Levels of mitochondrial protease CLPP were shown in original Fig. 1a, b. As a first step, we have conducted additional muscle autophagy and mitochondrial respiration measurements in muscles following denervation. We have found that, as predicted, the red mt-Keima fluorescence signal was significantly induced, an indicator of mitochondrial autophagy, in EDL muscles of mt-Keima Tg mice at 5 days following denervation (revised Supplementary Fig. 7a), which was accompanied by increased conversion of LC3-I to LC3-II and reduction of muscle mitochondrial respiration capacity (revised Supplementary Fig. 7b, c). Thus, our data suggest that denervation-induced muscle disuse leads to suppressed expression of LONP1 with increased muscle autophagy and

mitochondrial dysfunction, which is consistent with our observations in LONP1 mKO muscles. In addition, we have also added new data demonstrating that loss of LONP1 exacerbated denervation-induced muscle atrophy (revised Fig. 2b-e). Taken together, our data suggest that LONP1 acts as a component in the control circuitry during muscle disuse.

We think it is worth noting that according to published studies, the ubiquitin degradation system is activated during denervation-induced muscle loss (Bodine et al., *Science*. 2001;294(5547):1704-8; Gomes et al., *Proc Natl Acad Sci U S A*. 2001;98(25):14440-5). We also found increased expression of ubiquitin degradation genes (e.g. MAFbx and MuRF1) during the denervation process (revised Supplementary Fig. 7d). These data indicate that activation of the LONP1-dependent autophagy pathway, along with the ubiquitin degradation system, accompanies the process of disuse induced skeletal muscle loss.

The above information has been added as revised Fig. 2b-e and new Supplementary Fig. 7 (page 10, line 206-210, page 17, line 381-392) and addressed in the revised Discussion (page 25, line 565-568).

3. *“LonP1 is a crucial component of the proteostatic mechanism of mitochondria, it has been reported as an essential protease for the quality control of a number of mitochondrial systems, therefore it is not surprising that the ablation of this protein skeletal muscle can be deleterious. I am surprised by the absence of any evidence of activation of the ubiquitin-proteasome system when LonP1 is ablated, because this is a virtually universal mechanism to eliminate dysfunctional proteins, is partly linked to activation of autophagy, etc. Maybe this mechanism is not activated because most if not all changes in protein turnover induced by the ablation of LonP1 are affecting intramitochondrial proteins that are not processed by the ubiquitin-proteasome system? This is a possibility of course, but in the long-term the abolition of LonP1 should have some repercussion also on the general proteostasis of the cell, for instance by creating some effect on precursor proteins that should enter mitochondria through the outer and inner membrane-bound translocon proteins, etc. Therefore, evaluation of the absence of ubiquitination activation should be followed for a suitable amount of time.”*

Following the Reviewer’s suggestions, we examined the ubiquitin-proteasome system in muscles of LONP1 mKO mice at 16 weeks of age and observed similarly reduced expression of the ubiquitin degradation-related genes relative to WT controls (revised Supplementary Fig. 5b). Moreover, the content of ubiquitinated proteins in muscle homogenates from WT and LONP1 mKO mice was not different at 16 weeks of age (revised Supplementary Fig. 5c). Thus, the reviewer is correct, it is possible that LONP1 ablation mainly affects intramitochondrial proteins that are not processed by the ubiquitin-proteasome system. It is also possible that disturbed mitochondrial proteostasis triggers a unique mode of mitochondrial autophagy to selectively degrade mitochondrial domains. These new data have been added to the revised Supplementary Fig. 5b, c (page 14, line 299, 303-305) and this point has been slightly expanded in the revised Discussion (page 26, line 590-592).

4. *“The mitochondrial energy metabolism seems to be clearly affected, by the observation of both in vivo defects induced by ablation of LonP1 on muscle strength and endurance, and the evidence of reduced oxygen consumption under ADP and substrate administration. This result points to a defect in the proteostasis of the mitochondrial respiratory chain complexes. Although no changes in the amount of subunits of the respiratory chain complexes is reported in the paper, the Authors should*

further investigate the integrity of the quaternary structures of the respiratory complexes by BNGE-WB, with dodecyl-maltoside and, in order to evaluate the integrity of supercomplexes, digitonin as well. In addition, the activities of the single complexes should also be assessed, together with the measurement of ATP production and other bioenergetic features.”

We thank the reviewer. We followed the Reviewer’s suggestions, and we have improved our blue native polyacrylamide gel electrophoresis (BN-PAGE) analyses using new method (Wittig et al., Nat Protoc. 2006;1(1):418-28.) and replaced the original Supplementary Fig. 6 with the new results (revised Supplementary Fig. 4). The data now indicate a mild-to-moderate decrease in the amount of fully assembled complexes I, III, and IV in LONP1 mKO muscles relative to WT controls (revised Supplementary Fig. 4a). We also conducted in-gel activity assays (IGA) following BN-PAGE. Similarly, a mild decrease in in-gel activities of complexes I and IV were also observed in LONP1 mKO muscles (revised Supplementary Fig. 4b). These new data have been added to the revised Supplementary Fig. 4 (page 13, line 275-281, Methods, page 38, line 856-874).

5. *“The TEM results reported in the paper clearly show profound alterations of the structure and integrity of the organelles, which can well justify the induction of autophagy, or mitophagy, aimed at eliminating the defective mitochondria. The experiments using KEIMA are interesting and convincing, as it is the measurement of the autophagic flux by colchicine/chloroquine sequence. Nevertheless, this set of elegant experiments should be completed by some independent approaches, including the traditional measurement of autophagic markers, including LC3-II etc.”*

We thank the reviewer. As shown in original Fig. 4i, both basal and colchicine-induced accumulation of LC3-II protein were higher in LONP1 mKO muscles than WT controls. In addition, we have also included autophagy marker P62 to our autophagy flux analysis (revised Fig. 4i). These data support an enhanced autophagy flux in LONP1 mKO muscles.

6. *“The concordant possible increase of lysosomal biogenesis should be proven by measuring Lamp1 and TFEB (phosphorylated and non-phosphorylated).”*

We considered the Reviewer’s suggestions. Change in TFEB localization (shift between nucleus and cytoplasm) can reflect its phosphorylation/dephosphorylation status, we thus examined the localization of TFEB in LONP1 mKO muscles. We did not detect a change in TFEB nuclear translocation in muscles from LONP1 mKO mice based on standard immunohistochemistry studies (revised Supplementary Fig. 5i). In addition, we also found no difference in LAMP1 protein levels in LONP1 mKO muscles compared to WT controls (revised Supplementary Fig. 5h). These data suggest that LONP1 ablation might not activate lysosomal biogenesis in skeletal muscles. These new data have been added to the revised Supplementary Fig. 5h, i (page 15, line 334-339).

7. *“The profound alteration of the organelles associated with LonP1 ablation can indeed trigger autophagy/mitophagy. This is entirely reasonable, but the crucial question is whether this is a specific mechanism associated to LonP1 ablation, or rather a general autophagic response against*

any disruptive condition affecting so dramatically the shape, structure, and presumably function of mitochondria? This question is not really addressed in the paper, despite the experiments done with a truncated variant of the ornithine carbamylase presented at the end of the work, the meaning of it is not completely clear. Importantly, as mentioned above, it would be interesting to test whether the organelle wasting observed with a complete abolition of LonP1 is also present in the denervated or immobilized muscle associated with reduced muscle mass and amount of LonP1. The point concerning the specificity of the autophagic process induced by LonP1 is an essential issue which is not really addressed by the paper, since LonP1 acts on a myriad of different substrates and an autophagic control of general mitochondrial homeostasis is reasonably triggered by a drastic insult such as the complete elimination of this important proteostatic element of mitochondria.”

We appreciate the Reviewer’s concerns regarding whether our observed LONP1-directed regulation of autophagy program may be a general response against dramatic mitochondrial dysfunction, not a specific mechanism associated to LONP1 ablation. We have addressed this point experimentally and demonstrated a highly specific mechanism that link LONP1 deficiency to muscle cell autophagy.

We also think it is worth noting that muscle specific deletion of mitochondrial protease CLPP using muscle creatine kinase promoter driven Cre does not lead to a muscle phenotype compared to WT controls (Becker et al., EMBO Rep. 2018;19(5):e45126.). Our findings that loss of LONP1 impairs mitochondrial function and causes skeletal muscle loss and weakness revealed a specific physiological role of LONP1 in skeletal muscle.

We have conducted additional experiments and demonstrated that LONP1 regulates muscle cell autophagy through degradation of the PARK7 protein. In brief, we have conducted additional analyses of our unbiased iTRAQ proteomics data and tested possible LONP1 substrates involved in the regulation of autophagy. Interestingly, PARK7 protein, whose expression was upregulated in LONP1 mKO muscle mitochondria, was recently shown to regulate autophagy (Lee et al., Autophagy. 2018;14(11):1870-1885.; Xue et al., Arch Biochem Biophys. 2017;633:124-132; Krebiehl et al., PLoS One. 2010;5(2):e9367; Gao et al., J Mol Biol. 2012;423(2):232-48). We sought to determine whether PARK7 protein is an LONP1 substrate and involved in LONP1-mediated regulation of muscle cell autophagy. As a first step, we confirmed that PARK7 protein was dramatically upregulated (5-fold) in LONP1 KO myotubes compared to WT controls (revised Fig. 6e). However, PARK7 mRNA abundance was comparable in LONP1 KO myotubes versus WT controls (revised Fig. 6f), suggesting a post-transcriptional regulation of PARK7 protein. Second, we have blocked cytoplasmic protein synthesis (by cycloheximide treatment) in myotubes and followed the turnover of PARK7 protein. While in WT myotubes levels of PARK7 protein decreased over time, PARK7 protein was stabilized in the absence of LONP1 (revised Fig. 6g), these data provided further evidence that PARK7 is likely LONP1 substrate. Moreover, we found that PARK7 could directly interact with LONP1 based on standard co-immunoprecipitation studies in HEK293T cells (revised Fig. 6h). Together, these data suggest that PARK7 is a LONP1 substrate in muscle cells. To further determine whether PARK7 protein is involved in mediating LONP1 regulation of autophagy, the effects of siRNA-mediated knockdown of *Park7* were assessed in WT and LONP1 KO myotubes. We have found that siRNA knockdown of *Park7* reduced autophagy flux in LONP1 KO myotubes (revised Fig. 6i, j). Taken together, our mechanistic studies identified PARK7 as a LONP1 substrate,

which, when knocked down, reduced autophagy flux in LONP1-deficient muscle cells. While we cannot exclude the possibility that LONP1 may also act through other mechanisms to regulate autophagy, our findings identified a specific PARK7 mechanism that link LONP1 ablation to muscle cell autophagy. The above information has been added as revised Fig. 6e-j (page 19, line 422-442) and addressed in the revised Discussion (page 27, line 602-612).

We have also conducted additional mitochondria autophagy analysis in muscles following denervation as suggested. Please also see our response to point #2 above. We have found that, as predicted, the red mt-Keima fluorescence signal was significantly induced, an indicator of mitochondrial autophagy, in EDL muscles of mt-Keima Tg mice at 5 days following denervation (new Supplementary Fig. 7a), which is consistent with our observations in LONP1 mKO muscles. These new data have been added to new Supplementary Fig. 7a (page 17, line 381-385).

8. *“I am also wondering whether any effect towards a correction of the autophagic consequences of LonP1 ablation can be achieved by using antiautophagic agents such as brefeldin A, vincristine or chloroquine, and whether the morphological and biochemical alterations will persist under treatment with pro-autophagic agents such as rapamycin or rilmenidine.”*

We agree with the Reviewer in that it would be interesting to examine the effects of anti-autophagic agents on LONP1-mediated regulation of muscle function. Given that we found enhanced autophagy flux in LONP1 KO myotubes by using chloroquine (CQ) (Fig. 5l, m). The effects of autophagy inhibition using CQ were assessed in WT and LONP1 mKO mice. Analysis of fiber size distribution revealed that CQ treatment led to a reduced number of small fibers and an increased number of larger fibers in LONP1 mKO mice (revised Supplementary Fig. 5f, g). This supports a key role of autophagy in LONP1-mediated regulation of muscle function. These new data have been added to revised Supplementary Fig. 5f, g (page 15, line 331-334).

To further determine whether other autophagy signaling such as mTOR-dependent pathway is involved in LONP1-mediated regulation of muscle function, we explored the impact of rapamycin treatment in AAV-Cre mediated LONP1 knockout model. Rapamycin induced significant reduction of phosphorylated EIF4E-binding protein 1 (p-4EBP1), downstream product of mTORC1 activation, in both LONP1 knockout and control mice (new Supplementary Fig. 8a). Fiber area frequency distribution revealed that rapamycin treatment led to an increase in the percentage of small fibers (a leftward shift) in both LONP1 knockout and control mice (new Supplementary Fig. 8b, c). These results suggest an mTORC1-independent regulation of muscle function by LONP1. These new data have been added to new Supplementary Fig. 8 (page 17-18, line 392-399).

Itemized Responses to Reviewer #4

“This manuscript by Xu et al. seeks to demonstrate that disuse leads to loss of muscle LONP1, which is responsible for mitochondrial dysfunction resulting in reduced skeletal muscle mass and strength. The authors use a number of cellular and animal models in a variety of experimental systems to provide proof for this. They also include one experiment done on human muscle biopsies to extend the findings to human. I feel that although the majority of the experiments related to loss of LONP1 are well done, the overall evidence does not really substantiate their claims. I have the following concerns with the manuscript:”

We sincerely appreciate the Reviewer’s critical and constructive review of our manuscript.

Major issues:

1. *“The authors study samples obtained from the supraspinatus muscle from individuals diagnosed with rotator cuff injury. They claim that that the severity of this muscle loss depends upon the time from onset to the diagnosis. This is a really complicated system to use as there is not direct demonstration of muscle loss at all. In the early vs. late diagnosis groups, clinical scores that include a variety of measures including pain are provided and even those are highly overlapping across the groups. The location of the biopsy and the tissue recovered could skew the results greatly. Finally, the Western blots shown in Figure 1g are not convincing at all with differences observable only in two of the samples in the late diagnosis group - such small differences in light of the fact that Western blotting is not really a quantitative technique should be very carefully interpreted. I believe that the authors should obtain muscle biopsies from other, more extreme, clinical setting where the disuse and muscle loss is more clearly demonstrated.”*

We appreciate the Reviewer’s point. Although the severity of supraspinatus muscle loss is linked to the time from onset to the diagnosis. We agree with the Reviewer that the atrophy of the supraspinatus belly cannot be defined solely based on the duration of symptoms or clinical presentation. A previous study had reported a reproducible and reliable method to define supraspinatus muscle atrophy with the help of MRI (Thomazeau et al., Acta Orthop Scand. 1996;67(3):264-8). Specifically, by calculating the occupation ratio which is the ratio between the surface of the cross-section of the muscle belly and that of the supraspinatus fossa, it allowed a reliable measurement of supraspinatus muscle atrophy and the muscle with a ratio greater than 0.60 can be considered as normal or non-atrophied, while values below 0.60 suggest muscle atrophy.

We have now evaluated the atrophy of the supraspinatus muscle using the MRI-based method (Thomazeau et al., Acta Orthop Scand. 1996;67(3):264-8). Briefly, MRI studies were carried out on 9 new patients, and we calculated the occupation ratio of the supraspinatus fossa by the muscle belly (see Figure 1 below). Based on the occupation ratio and previous report (Thomazeau et al., Acta Orthop Scand. 1996;67(3):264-8), we divided the patients into two groups (non-atrophy group, mean ratio 0.76; atrophy group, mean ratio 0.50). The data now indicate a clear muscle loss in atrophy group (revised Supplementary Table 1). We found that the protein levels of LONP1 and CLPP were significantly reduced in human muscles from the atrophy group compared to non-atrophy group (revised Fig. 1g, h), In addition, both LONP1 and CLPP proteins exhibited a significant positive correlation with occupation ratios

(revised Fig. 1i). In contrast, the levels of OPA1 protein were not different between atrophy and non-atrophy groups, and neither exhibited a significant correlation with occupation ratios (revised Fig. 1g and revised Supplementary Fig. 1f, g). Our new data suggest that muscle loss in humans is associated with decreased mitochondrial proteases. The new data have been added to the revised Fig. 1g-i and revised Supplementary Fig. 1f, g (page 8-9, line 159-173).

Notably, all the supraspinatus muscle biopsies were taken at the same area (2 cm proximal zone of muscle tendon junction) under direct visualization from the arthroscope during rotator cuff repair surgery. We have added more details in regards to human supraspinatus muscle studies in the Methods section (page 30-31, line 662-689).

2. *“The authors have done a proteomics experiments using iTRAQ-based labeling for quantitation. They describe that they considered proteins that displayed 1.2 fold-change.*
 - a. *These experiments were done in duplicate in mice. I wonder why triplicate measurements were not done so that the authors could do robust statistical analyses. Also, the distribution of changes (such as in a waterfall plot) are not shown.*
 - b. *Their cutoff of 20% change is also highly arbitrary and a really low cutoff to be able to make confident conclusions. Even across the replicate mice, their results shown in a highly diagrammatic fashion in Figure 6d are not really correlated. Since they describe ornithine transcarbamylase (OTC) as a substrate of LONP1, I do not see OTC being mentioned as a molecule whose level was altered in this system.*
 - c. *They do not discuss if they saw a single peptide or multiple peptides to derive their protein identification and quantitation. If multiple peptides were detected, what was the distribution of fold-changes across different peptides from the same protein?*
 - d. *Finally, they should deposit their proteomics data to a public repository such as PRIDE.”*

We think the Reviewer has raised an important point and we did not present our data clearly. We have performed additional analysis of our iTRAQ proteomics data and provided additional experimental data to make our data clearer.

- a) We agree with the reviewer that an increase in sample size will help in this study. Unfortunately, because of the grant budget and the limited amount of mitochondria purified from 2-week-old mice muscles, we used an iTRAQ Reagent-4 plex kit, which is suitable for labeling total four samples (two independent mitochondria samples from WT muscles and two independent mitochondria samples from LONP1 mKO muscles). We have provided more details with regard to iTRAQ sample labeling in the Methods section (page 35, line 792-799). As suggested, we have also added a waterfall plot in new Supplementary Fig. 9a for the mitochondrial proteins identified in our proteomics experiment.
- b) We agree with the reviewer that 1.2 fold-change cut-off is not very stringent. The reason we use fold-change > 1.2 as threshold is that we found iTRAQ usually reduces the real changes/differences between different samples. For instance, a protein with fold-change > 1.2 quantified by iTRAQ can be fold-change > 2-3 when determined by Western blot. Moreover, fold-change > 1.2 as cut-off is also a generally accepted threshold which has been used in many iTRAQ proteomics studies (Zhang et al., *J Proteome Res.* 2020;19(4):1788-1799; Cao et al., *J Proteome Res.* 2019;18(5):2032-2044; Singh et al., *J Proteomics.* 2021;236:104125; Cheng et al., *Oncol Lett.* 2017;14(6):8084-8091). The data in original Fig. 6d showed the similarity between the LONP1 mKO duplicate mitochondria samples. There are unavoidable within group variations for mouse experiments, which account for these results. We have performed additional analyses and modified the Fig. 6d in order to more accurately reflect these results (revised Fig. 6d). Specifically, we have also included the duplicate WT samples to the heat-map analysis for better clarity. Moreover, we further performed analysis to identify the differences between the KO group and the WT group. As shown in the following PCA and sample distance plots (see Figure 2 below), the variations between the biological replicates within each group are smaller than the difference between the KO group and the WT group. Therefore, the KO should be the primary factor which is responsible for the observed changes. Importantly, our findings of LONP1 regulation of mitochondrial protein identified from these unbiased proteomics analyses have been validated by our functional experimental assays. Specifically, we confirmed that autophagy protein PARK7, whose expression was upregulated in LONP1 mKO muscle mitochondria (original Fig. 6d), was markedly increased (5-fold) in LONP1 KO myotubes (revised Fig. 6e). In the absence of LONP1, PARK7 protein was stabilized (revised Fig. 6g). In addition, our new mechanistic studies identified PARK7 as a LONP1 substrate, which, when knocked down, reduced autophagy flux in LONP1-deficient muscle cells (revised Fig. 6i, j). Therefore, our new experimental data support our mass spectrometry proteomics analysis.
The deletion mutant of ornithine transcarbamylase (Δ OTC) was addressed in the point #3 below.
- c) The original MS and MS/MS data of our iTRAQ experiment was searched using ProteinPilot Software (version 4.5, AB Sciex) with Paragon Algorithm. If multiple peptides were detected, the software will automatically select the peptides fitting in its selection criteria for protein quantification, most likely using their average to generate fold change information among different samples. We have provided more details with regard to MS data analysis in the Methods section (page 36-37, line 812-813, 819-828).

- d) As suggested, we have deposited our proteomics data to the ProteomeXchange Consortium via the PRIDE partner repository with the dataset identifier PXD029722. Submission details: Project Name, iTRAQ-based analysis of muscle mitochondria isolated from LONP1-mKO mice; Project accession, PXD029722; Project DOI, Not applicable. Reviewer account details: Username, reviewer_pxd029722@ebi.ac.uk; Password, sqhFJ9CQ. This information has been added to revised Data availability (page 45, line 1035-1041).

3. *“I do not think overexpression experiments of a mitochondrial-localized mutant OTC are directly applicable to the role of LONP1. This is a highly artificial experiment, which only proves that overexpression of OTC in mitochondria in muscle causes muscle loss and not necessarily any connection to LONP1.”*

The Reviewer has raised an important point regarding Δ OTC overexpression in skeletal muscle. We think it is worth noting that the deletion mutant of ornithine transcarbamylase (Δ OTC) is a known protein degraded by LONP1 and a well-established model for studying mitochondrial proteostasis imbalance (Zhao et al., EMBO J. 2002;21(17):4411-9; Jin SM, Youle RJ. Autophagy. 2013;9(11):1750-7; Bezawork-Geleta et al., Sci Rep. 2015;5:17397; Quirós et al., Nat Rev Mol Cell Biol. 2015;16(6):345-59.). In fact, the physiological relevance of imbalanced mitochondrial proteostasis remains largely unclear. We showed that muscle-specific overexpressing mitochondrial-retained Δ OTC protein is sufficient to induce mitochondrial dysfunction, autophagy activation, and cause muscle loss and weakness, thus, we suggest that the accumulation of misfolded proteins in mitochondria is a physiological trigger of mitochondrial dysfunction and muscle loss is an important and novel aspect in this regard.

We have also conducted additional experiments to further support the connection between LONP1 and Δ OTC. Notably, it has been established that increased workload of unfolded proteins at the mitochondrial can trigger the activation of the mitochondrial proteases. Consistently, we have found that overexpression of Δ OTC triggered a mild induction of LONP1 protein in skeletal muscles (new Supplementary Fig. 9b), indicating a mitochondrial proteostasis stress in MCK- Δ OTC muscles. We also

examined whether overexpression of LONP1 could impact the muscle phenotypes in MCK- Δ OTC mice. We generated recombinant AAV2/9 to express LONP1 in MCK- Δ OTC muscle *in vivo* through systemic delivery to neonatal mice. Consistent with LONP1 degrading Δ OTC (Jin SM, Youle RJ. *Autophagy*. 2013;9(11):1750-7; Bezawork-Geleta et al., *Sci Rep*. 2015;5:17397), overexpression of LONP1 resulted in a significant reduction of Δ OTC protein in MCK- Δ OTC muscles (new Fig. 8b). As predicted, we have found that overexpression of LONP1 not only reduced the amount of LC3-II, but also significantly increased the mitochondrial respiration capacity in MCK- Δ OTC muscles (new Fig. 8c, d). Moreover, quantification of muscle fiber size distribution revealed that overexpression of LONP1 also led to a shift toward larger myofibers in MCK- Δ OTC mice (new Fig. 8e, f). These results further suggest that LONP1 play a crucial role in preserving muscle function through maintaining mitochondrial proteostasis.

The above information has been added as new Fig. 8a-f and new Supplementary Fig. 9b (page 20, line 452-454, page 21-22, line 485-496) and addressed in the revised Discussion (page 24, line 538-545).

Minor issues:

1. *“The authors refer to an impact on mitochondrial “quality” - I think this phrase should be avoided and they should restrict their discussion to mitochondrial dysfunction or other more precise descriptions.”*

We thank the reviewer. We have changed the term “mitochondrial quality” to “mitochondrial function” as suggested.

Reviewers' Comments:

Reviewer #1:

Remarks to the Author:

the authors responded satisfactorily to the doubts I raised, and added some data regarding mitophagy - in particular, the identification of PARK7 as a possible substrate of Lonp1 in muscle - which added further value and meaning to the manuscript.

I have no further comments or requests to the authors.

Reviewer #2:

Remarks to the Author:

The authors have, in this revised manuscript, satisfactorily addressed all of my original concerns. In doing so, the data provided support the conclusions set forth by the authors.

Reviewer #3:

Remarks to the Author:

I am altogether happy for the changes introduced in the presentation of results and text concerning my own critiques. The paper is interesting and its science sound and novel.

Reviewer #4:

Remarks to the Author:

The authors have addressed all of my previous concerns and I feel that the revised manuscript is now suitable for publication.

Itemized Responses to Reviewer #1

“The authors responded satisfactorily to the doubts I raised, and added some data regarding mitophagy - in particular, the identification of PARK7 as a possible substrate of Lonp1 in muscle - which added further value and meaning to the manuscript. I have no further comments or requests to the authors.”

We thank this reviewer for a comprehensive and insightful review.

Itemized Responses to Reviewer #2

“The authors have, in this revised manuscript, satisfactorily addressed all of my original concerns. In doing so, the data provided support the conclusions set forth by the authors.”

We thank this reviewer for a comprehensive and insightful review.

Itemized Responses to Reviewer #3

“I am altogether happy for the changes introduced in the presentation of results and text concerning my own critiques. The paper is interesting and its science sound and novel.”

We thank this reviewer for a comprehensive and insightful review.

Itemized Responses to Reviewer #4

“The authors have addressed all of my previous concerns and I feel that the revised manuscript is now suitable for publication.”

We thank this reviewer for a comprehensive and insightful review.